# Concrete Score Matching: Generalized Score Matching for Discrete Data

**Chenlin Meng**[*]
Stanford University
chenlin@cs.stanford.edu

**Kristy Choi**[*]
Stanford University
kechoi@cs.stanford.edu

**Jiaming Song**
NVIDIA
jiamings@nvidia.com

**Stefano Ermon**
Stanford University, CZ Biohub
ermon@cs.stanford.edu

## Abstract

Representing probability distributions by the gradient of their density functions has proven effective in modeling a wide range of continuous data modalities. However, this representation is not applicable in discrete domains where the gradient is undefined. To this end, we propose an analogous score function called the "Concrete score", a generalization of the (Stein) score for discrete settings. Given a predefined neighborhood structure, the Concrete score of any input is defined by the rate of change of the probabilities with respect to local directional changes of the input. This formulation allows us to recover the (Stein) score in continuous domains when measuring such changes by the Euclidean distance, while using the Manhattan distance leads to our novel score function in discrete domains. Finally, we introduce a new framework to learn such scores from samples called Concrete Score Matching (CSM), and propose an efficient training objective to scale our approach to high dimensions. Empirically, we demonstrate the efficacy of CSM on density estimation tasks on a mixture of synthetic, tabular, and high-dimensional image datasets, and demonstrate that it performs favorably relative to existing baselines for modeling discrete data.

## 1 Introduction

When estimating a statistical model, the representation of the underlying probability distribution has profound implications on the downstream modeling task. Likelihood-based model families such as Variational Autoencoders (VAEs) [1–4], normalizing flows [5–13], and diffusion models [14–16] are forced to either approximate the intractable normalizing constant or utilize restrictive model architectures; implicit models [17–20] try to capture the underlying sampling process by relying on unstable adversarial training procedures. On the other hand, representing the distribution as the gradient of the log probability density function—also known as the (Stein) score—allows us to circumvent such issues. This is key to score matching's success on a wide range of continuous data modalities [21–25]. In fact, its recent resurgence has led to significant advances in machine learning applications of density estimation, such as image generation [26, 16, 27, 15] and audio synthesis [28, 29], among others.

However, score matching approaches that rely on modeling the gradient of the data distribution are inherently designed for continuous data; such methods hinge on the *existence* of the gradient, which is undefined for discrete domains [22]. Given the ubiquitous nature of structured, discrete data in

---

[*]Joint first author

our world today such as graphs, text, genomic sequences, images, we desire an approach that would allow us to expand the successes of score-based generative models into discrete domains.

To address this challenge, we first introduce the Concrete score: a generalization of the (Stein) score that is amenable to both continuous and discrete data types[2]. Given a predefined neighborhood structure, the Concrete score leverages structural information in the data to construct *surrogate gradient information* about the discrete space by considering the similarity between two neighboring examples. More precisely, the Concrete score of any input is defined by the rate of change of the probabilities with respect to directional changes of the input. In direct analogy to the continuous (Stein) score, the intuition is that the Concrete score should remain small when two examples are "close" to one another; when the two examples are very different, the Concrete score will be large. We find that measuring such changes by the Euclidean distance recovers the (Stein) score in continuous domains, while using the Manhattan distance leads to our novel score function in discrete domains.

Using this definition of the Concrete score, we then introduce a framework to learn such scores from samples called Concrete Score Matching (CSM). To successfully scale our method to high dimensions, we also propose an efficient training objective as well as several variations of our approach that improve performance in practice. Empirically, we demonstrate the efficacy of CSM on a wide range of density estimation tasks on a mixture of synthetic, tabular, and high-dimensional image datasets, and demonstrate that it performs favorably relative to existing baselines for modeling discrete data.

The contributions of our work can be summarized as follows:

1. We propose the Concrete score, a generalization of the (Stein) score to discrete domains. We construct the Concrete score such that it leverages structural information in the data to capture surrogate "gradient" information over discrete spaces.

2. We introduce a novel score matching framework for estimating such scores from samples called Concrete Score Matching (CSM). We then outline several connections between CSM and existing (continuous) score matching techniques, such denoising score matching (DSM).

3. We propose efficient learning objectives that allow our method to scale gracefully to high-dimensional datasets, and show how to port over the recent successes in continuous score matching approaches to discrete domains.

## 2 Preliminaries

Let $p_{\text{data}}(\mathbf{x})$ be the unknown data distribution over $\mathcal{X} \in \mathbb{R}^D$, for which we have access to i.i.d. samples $\{\mathbf{x}_i\}_{i=1}^N \sim p_{\text{data}}(\mathbf{x})$. The (Stein) score of $p_{\text{data}}(\mathbf{x})$ is the first order derivative of the log data density function $\boldsymbol{s}(\mathbf{x}) = \nabla_{\mathbf{x}} \log p(\mathbf{x})$. The goal of score matching is to learn an unnormalized density $q_\theta(\mathbf{x})$ indexed by $\theta \in \Theta$ such that the estimated score function $\boldsymbol{s}_\theta(\mathbf{x}) = \nabla_{\mathbf{x}} \log q_\theta(\mathbf{x})$ is close to $\boldsymbol{s}(\mathbf{x})$.

In practice, we leverage a *score network* $\boldsymbol{s}_\theta : \mathbb{R}^D \to \mathbb{R}^D$ that is trained to minimize the Fisher divergence between $q_\theta(\mathbf{x})$ and $p_{\text{data}}(\mathbf{x})$ [23, 26]. Although the original objective is intractable to compute due to its dependence on the ground truth data scores $\boldsymbol{s}(\mathbf{x})$, we can leverage integration by parts [21] to simplify the objective as follows:

$$\mathcal{J}_{SM}(\theta) = \mathbb{E}_{p_{\text{data}}(\mathbf{x})}\left[\frac{1}{2}\|\boldsymbol{s}_\theta(\mathbf{x})\|_2^2 + \text{tr}(\nabla_{\mathbf{x}} \boldsymbol{s}_\theta(\mathbf{x}))\right] + \text{const.} \tag{1}$$

where $\text{tr}(\cdot)$ denotes the matrix trace, and the optimal score model satisfies $\boldsymbol{s}_{\theta^*}(\mathbf{x}) \approx \boldsymbol{s}(\mathbf{x})$. While the trace term in Eq. 1 remains problematic—it requires an expensive evaluation of the Hessian of the log-density function—recent works [23, 16] leverage the Skilling-Hutchinson trace estimator [31, 32] or directional derivatives [33] to efficiently approximate the training objective.

Despite their key role in successfully scaling score-based generative models to high-dimensional datasets, we note two critical limitations of existing score matching techniques: (1) $\mathcal{X}$ must be continuous; and (2) the density function $p_{\text{data}}(\mathbf{x})$ must be differentiable. This poses a significant challenge in discrete domains, where both requirements fail to hold.

---

[2]We borrow the terminology from "concrete mathematics" [30].

# 3 The Concrete Score

The above limitations prevent the direct application of score matching techniques to discrete data. We thus propose the Concrete score, a generalization of the (Stein) score for discrete settings. The key intuition is that although the gradient is undefined in discrete spaces, we can still construct a *surrogate* for the gradient by leveraging local directional changes to the input. We do so by exploiting special neighborhood structures in the data, and elaborate upon the necessary conditions on these structures to guarantee that our surrogate gradient indeed recovers a valid score function that (1) completely characterizes the distribution, and (2) is amenable for parameter estimation.

## 3.1 Constructing Surrogate Gradients for Discrete Data

To be more precise, let $p_{\text{data}}(\mathbf{x})$ be the data distribution over $\mathcal{X}$. We denote $\mathcal{N} : \mathcal{X} \to \mathcal{X}^K$ as the function mapping each example $\mathbf{x} \in \mathcal{X}$ to a set of neighbors, such that $\mathcal{N}(\mathbf{x}) = \{\mathbf{x}_{n_1}, ..., \mathbf{x}_{n_k}\}$ and $K = |\mathcal{N}(\mathbf{x})|$ [3]. This neighborhood induces a particular graphical structure onto the support of $p_{\text{data}}(\mathbf{x})$, which we call the "neighborhood-induced graph", that will play a key role in constructing the surrogate gradient. We provide a formal definition below.

**Definition 1** (Neighborhood-induced graph). *Let $p_{\text{data}}(\mathbf{x})$ be the data distribution over $\mathcal{X}$ and $\mathcal{N}$ be the function mapping each node $\mathbf{x} \in \mathcal{X}$ to its set of neighbors. The neighborhood-induced graph $\mathcal{G}$ is the directed graph which results from adding a directed edge from $\mathbf{x}$ to each node in its neighborhood set $\mathbf{x}_n \in \mathcal{N}(\mathbf{x})$, for all $\mathbf{x} \in \text{supp}(p_{\text{data}}(\mathbf{x}))$.*

An important point is that the neighborhood structure can be asymmetric, since the neighborhood-induced graph is a directed graph. This implies that there may exist cases where $\mathcal{N}(\mathbf{x}) = \{\mathbf{x}_1\}$ does not necessarily imply that $\mathcal{N}(\mathbf{x}_1) = \{\mathbf{x}\}$. We provide an intuitive example below.

**Example 1.** *When $\mathcal{X} = \{\mathbf{x}_0, \mathbf{x}_1, \ldots, \mathbf{x}_5\}$, and the neighborhood structure $\mathcal{N}$ is defined as: $\mathcal{N}(\mathbf{x}_0) = \{\emptyset\}$ and $\mathcal{N}(\mathbf{x}_i) = \{\mathbf{x}_0\} \ \forall i = 1, \ldots, 5$, the neighborhood-induced graph is the star graph in Figure 1.*

There exist a wide range of neighborhood structures, all of which yield different neighborhood-induced graphs. We visualize five common structures in Figure 1.

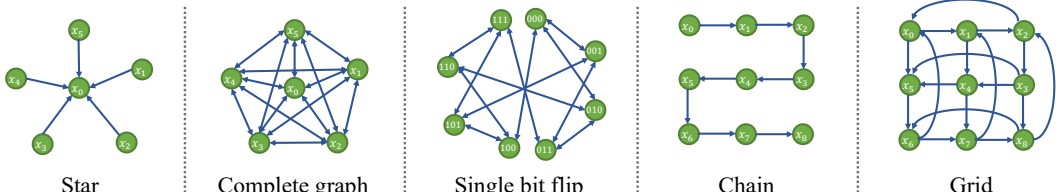

| Star | Complete graph | Single bit flip | Chain | Grid |

Figure 1: Examples of common neighborhood structures and their corresponding (connected) neighborhood-induced graphs, where the arrows point to the neighbors of a given node. Note that the neighborhood structure is directed.

Such graphs provide insight into the kinds of neighborhood structures that are amenable to our framework. One necessary characteristic of such graphs is *connectedness*. In fact, the five common graphical structures (among others) shown in Figure 1 are all weakly connected graphs. We emphasize that connectedness does not take the directionality of the underlying graph $\mathcal{G}$'s edges into account—it does not matter whether a node $\mathbf{x}$ has an incoming or outgoing edge.

With the above definitions in place, we can now define the surrogate gradient as the local differences between a set of examples (similar to directional derivative in the continuous case). Using this notion of the "gradient", we construct the Concrete score as the rate of change of the probabilities with respect to these local directional changes in the input $\mathbf{x}$ as defined below.

**Definition 2** (Concrete score). *Let $\mathcal{N}$ be a function mapping each point $\mathbf{x}$ to its set of neighbors $\mathcal{N}(\mathbf{x}) = \{\mathbf{x}_{n_1}, ..., \mathbf{x}_{n_k}\}$. Then, the Concrete score $\mathbf{c}_{p_{\text{data}}}(\mathbf{x}; \mathcal{N}) : \mathcal{X} \to \mathbb{R}^{|\mathcal{N}(\mathbf{x})|}$ for a given distribution $p_{\text{data}}(\mathbf{x})$ evaluated at $\mathbf{x}$ is:*

$$\mathbf{c}_{p_{\text{data}}}(\mathbf{x}; \mathcal{N}) \triangleq \left[ \frac{p_{\text{data}}(\mathbf{x}_{n_1}) - p_{\text{data}}(\mathbf{x})}{p_{\text{data}}(\mathbf{x})}, ..., \frac{p_{\text{data}}(\mathbf{x}_{n_k}) - p_{\text{data}}(\mathbf{x})}{p_{\text{data}}(\mathbf{x})} \right]^T. \tag{2}$$

---

[3] We consider a fixed number of neighbors $K$ for each $\mathbf{x}$ to simplify our notation, but note that this can be generalized to a variable number of neighbors for every $\mathbf{x}$.

Although we have defined the Concrete score, we have yet to justify whether it is indeed a suitable score function for estimating $p_{\text{data}}(\mathbf{x})$. The necessary condition, which we call *completeness* [34], ensures that the Concrete score preserves enough information about $p_{\text{data}}(\mathbf{x})$ such that we can successfully learn $p_{\text{data}}(\mathbf{x})$ from data samples using score matching. We make this statement more precise in the following theorem.

**Theorem 1** (Completeness). *Let $p_{\text{data}}(\mathbf{x})$ be a (discrete) data distribution. Denote $\mathbf{c}_{p_{\text{data}}}(\mathbf{x}; \mathcal{N})$ as the Concrete score of $p_{\text{data}}(\mathbf{x})$ with neighboring structure $\mathcal{N}$, and $\mathbf{c}_\theta(\mathbf{x}; \mathcal{N})$ the Concrete score for a distribution $p_\theta(\mathbf{x})$ parameterized by $\theta \in \Theta$. When the graph induced by the neighborhood structure $\mathcal{N}$ is connected, $c_\theta(\mathbf{x}; \mathcal{N}) = \mathbf{c}_{p_{\text{data}}}(\mathbf{x}; \mathcal{N})$ implies that $p_\theta(\mathbf{x}) = p_{\text{data}}(\mathbf{x}) \; \forall \mathbf{x} \in \mathcal{X}$.*

*Proof sketch.* If two nodes $\mathbf{x}$ and $\mathbf{x}'$ in a neighborhood-induced graph $\mathcal{G}$ share an edge, then their density ratio $p_{\text{data}}(\mathbf{x}')/p_{\text{data}}(\mathbf{x})$ can be uniquely identified using $\mathbf{c}_{p_{\text{data}}}(\mathbf{x}; \mathcal{N})$. Thus when $\mathcal{G}$ is connected, the density ratio between any node pairs in $\mathcal{G}$ is uniquely identified given $\mathbf{c}_{p_{\text{data}}}(\mathbf{x}; \mathcal{N})$. Therefore, knowing the density ratio between any two points uniquely identifies $p_{\text{data}}(\mathbf{x})$. $\square$

We provide the full proof in Appendix A. We emphasize here that the connectedness of the neighborhood-induced graph (a kind of regularity condition on $\mathcal{N}$ and $p_{\text{data}}(\mathbf{x})$) was crucial for demonstrating the completness of the Concrete score.

Note that Theorem 1 only requires the neighborhood-induced graph to be connected. This implies that given a connected graph, we can augment it with additional edges—the new graph will still remain complete, but contain additional information in the augmented neighborhood structure that may help parameter estimation in practice. This is a phenomenon that we observe empirically in Section 5.3.

## 3.2 Connection to Stein Scores and Existing Score Matching Techniques

The form of the Concrete score in Eq. 2 suggests a natural connection with the Stein score in continuous domains. In the following proposition, we illustrate the connection between the two when we define the neighborhood structure to be similar to the grid in Figure 1. In particular, as the distance between neighboring nodes shrinks to zero, our Concrete score converges to the Stein score up to a multiplicative constant.

**Proposition 1.** *For $\mathbf{x} \in \mathbb{R}^D$, $p(\mathbf{x}) \in \mathcal{P}(\mathbb{R}^D)$, and $\delta > 0$, let the neighborhood structure $\mathcal{N}_\delta(\mathbf{x}) = \{\mathbf{x} + \delta \mathbf{e}_i\}_{i=1}^D$. Then, we have:*

$$\lim_{\delta \to 0} \frac{\mathbf{c}_p(\mathbf{x}; \mathcal{N}_\delta)}{\delta} = \nabla_\mathbf{x} \log p(\mathbf{x}).$$

We note that while from this perspective the Concrete score can be seen as a finite-difference approximation to the continuous (Stein) score, CSM is different from finite difference score matching (FD-SM) [33]. We introduce a new family of score functions generalizable to discrete domains in the form of $(p(\mathbf{x} + \mathbf{v}) - p(\mathbf{x}))/p(\mathbf{x})$, where $\mathbf{v}$ is the direction from $\mathbf{x}$ pointing to its neighbor with length $\delta$. On the other hand, FD-SM still attempts to match the Stein score, and approximates directional derivatives with finite difference using two neighbors in the form of $\log p(\mathbf{x} + \mathbf{v}) - \log p(\mathbf{x} - \mathbf{v})$. Nevertheless, the differences between the two forms are $o(\delta)$ as $\delta \to 0$.

## 3.3 Inference with Concrete Scores

To perform inference, we note that there exist several ways to define Markov chain Monte Carlo (MCMC) samplers that only rely on the Concrete score—we highlight the simplest setting of Metropolis-Hastings [35–37] for clarity of exposition. We first note that, by definition:

$$\mathbf{c}_{p_{\text{data}}}(\mathbf{x}; \mathcal{N}) + \mathbf{1} = \left[ \frac{p_{\text{data}}(\mathbf{x}_{n_1})}{p_{\text{data}}(\mathbf{x})}, ..., \frac{p_{\text{data}}(\mathbf{x}_{n_k})}{p_{\text{data}}(\mathbf{x})} \right]^T \tag{3}$$

This implies that we can obtain the density ratios between the neighboring pairs $p_{\text{data}}(\mathbf{x}_{n_1})/p_{\text{data}}(\mathbf{x})$ and $p_{\text{data}}(\mathbf{x})/p_{\text{data}}(\mathbf{x}_{n_1})$ given the Concrete score. Given a proposal distribution $q$, our sampler will accept the proposed update $\mathbf{x}'$ with acceptance probability $A(\mathbf{x}'|\mathbf{x}) = \min\left(1, \frac{p_{\text{data}}(\mathbf{x}')q(\mathbf{x}|\mathbf{x}')}{p_{\text{data}}(\mathbf{x})q(\mathbf{x}'|\mathbf{x})}\right)$. Specifically, we can choose $q$ to sample uniformly among $\mathcal{N}(\mathbf{x})$ given a particular $\mathbf{x}$. When the neighborhood structure is symmetric, this selection leads to a simplified acceptance probability $A(\mathbf{x}'|\mathbf{x}) =$

$\min\left(1, \frac{p_{\text{data}}(\mathbf{x}')}{p_{\text{data}}(\mathbf{x})}\right)$. We note that when the underlying graph is connected, the chain is aperiodic (due to the presence of a rejection step), irreducible, and positive recurrent (since there is a finite number of discrete states), so the Markov chain is guaranteed to converge in the limit to the model distribution.

This means that we can also compute tighter bounds on log-likelihood estimates obtained by unnormalized probability models trained via CSM. In particular, we can compute both a lower bound estimated via Annealed Importance Sampling (AIS) [38] and a conservative upper bound estimated via the Reverse Annealed Importance Sampling Estimator (RAISE) [39]. This is because both AIS and RAISE allow us to approximate the intractable normalizing constant by constructing a sequence of intermediate distributions between our estimated target distribution and another proposal distribution, and we know that the Concrete score can be repurposed to obtain the necessary density ratios between neighboring pairs of distributions along this sequence.

## 4 Learning Concrete Scores with Concrete Score Matching

### 4.1 The Concrete Score Matching Objective

Next, under the assumption that $\mathcal{N}$ induces a weakly connected graph over the support of $p_{\text{data}}(\mathbf{x})$, we propose a new score matching framework called Concrete Score Matching (CSM) for estimating Concrete scores from data samples. To do so, we estimate $\boldsymbol{c}_{p_{\text{data}}}(\mathbf{x}; \mathcal{N})$ with a score model $\boldsymbol{c}_\theta(\cdot; \mathcal{N})$ : $\mathbb{R}^D \to \mathbb{R}^{|\mathcal{N}(\mathbf{x})|}$, where $|\mathcal{N}(\mathbf{x})|$ denotes the size of the neighborhood of $\mathbf{x}$. Our proposed training objective is quite natural, as it measures the average $\ell_2$ difference between our score model $\boldsymbol{c}_\theta(\cdot; \mathcal{N})$ and the true score $\boldsymbol{c}_{p_{\text{data}}}(\cdot; \mathcal{N})$ similar to (continuous) score matching:

$$\mathcal{L}_{\text{CSM}}(\theta) = \sum_{\mathbf{x}} p_{\text{data}}(\mathbf{x}) \left\| \boldsymbol{c}_\theta(\mathbf{x}; \mathcal{N}) - \boldsymbol{c}_{p_{\text{data}}}(\mathbf{x}; \mathcal{N}) \right\|_2^2 \tag{4}$$

In the following theorem, we demonstrate that Eq. 4 is indeed a principled learning objective: the estimated $\theta^*$ is a consistent estimator for the true underlying distribution $p_{\text{data}}(\mathbf{x})$.

**Theorem 2** (Consistency). *In the limit of infinite data and infinite model capacity, the optimal $\theta^*$ that minimizes Eq. 4 recovers the true Concrete score, or satisfies $\boldsymbol{c}_{\theta^*}(\mathbf{x}; \mathcal{N}) = \boldsymbol{c}_{p_{\text{data}}}(\mathbf{x}; \mathcal{N})$.*

Theorem 1 implies that the estimated (underlying) $p_{\theta^*}(\mathbf{x})$ from $c_{\theta^*}(\mathbf{x}; \mathcal{N})$ satisfies $p_{\theta^*}(\mathbf{x}) = p_{\text{data}}(\mathbf{x}) \ \forall \mathbf{x} \in \mathcal{X}$. Next, we note that Eq. 4 can be simplified to an objective that we can tractably optimize by removing its dependence on the unknown $\boldsymbol{c}_{p_{\text{data}}}(\mathbf{x}; \mathcal{N})$, similar in spirit to Eq. 1. Although the added flexibility of the score model may potentially lead to situations where it does not correspond to a valid probability distribution—similar to how continuous score models do not necessarily form a conservative vector field—this does not appear to hurt CSM's performance empirically.

**Theorem 3.** *Optimizing Equation* (4) *is equivalent to optimizing:*

$$\mathcal{J}_{\text{CSM}}(\theta) = \underbrace{\sum_{\mathbf{x}} \sum_{i=1}^{|\mathcal{N}(\mathbf{x})|} p_{\text{data}}(\mathbf{x}) \left( \boldsymbol{c}_\theta(\mathbf{x}; \mathcal{N})_i^2 + 2\boldsymbol{c}_\theta(\mathbf{x}; \mathcal{N})_i \right)}_{\mathcal{J}_1} - \underbrace{\sum_{\mathbf{x}} \sum_{i=1}^{|\mathcal{N}(\mathbf{x})|} 2p_{\text{data}}(\mathbf{x}_{n_i}) \boldsymbol{c}_\theta(\mathbf{x}; \mathcal{N})_i}_{\mathcal{J}_2}$$
$$\tag{5}$$

*where $\mathcal{N}(\mathbf{x}) = \{\mathbf{x}_{n_1}, ..., \mathbf{x}_{n_k}\}$ is the set of neighbors of $\mathbf{x}$.*

We note that in Eq. 5, the objective $\mathcal{J}_{\text{CSM}}(\theta)$ can be approximated using Monte Carlo samples. We elaborate on the empirical training details in Section 4.2.

### 4.2 Efficient Training via Monte Carlo

In practice, Eq. 5 is still inefficient; it involves a summation over all $|\mathcal{N}(\mathbf{x})|$ neighbors for $\mathbf{x}$, which can be expensive for high-dimensional datasets. Therefore, we propose several modifications to the original training objective that we found to be crucial for scaling up and improving our approach.

We begin with the leftmost term $\mathcal{J}_1$. First, we approximate the outer expectation w.r.t. $p_{\text{data}}(\mathbf{x})$ with Monte Carlo samples from the empirical data distribution. Next, rather than evaluating the objective for all $|\mathcal{N}(\mathbf{x})|$ neighbors, we sample a neighbor $\mathbf{x}_{n_i}$ uniformly at random among $\mathcal{N}(\mathbf{x}_i)$ for

every $\mathbf{x}_i$ in a mini-batch. We then upweight the output from the model using this sample by $|\mathcal{N}(\mathbf{x})|$. We provide the unbiased training algorithm for the leftmost term in Algorithm 1. We note that is similar in spirit to sliced score matching [23], an approximation technique to make continuous score matching scalable to higher dimensions.

Next, we turn to the rightmost term $\mathcal{J}_2$. Similar to $\mathcal{J}_1$, we can estimate $\mathcal{J}_2$ using an unbiased estimator based on samples as demonstrated in Algorithm 2. We define $\mathcal{N}^{-1}(\mathbf{x}') = \{(\mathbf{x}, i)|\mathcal{N}(\mathbf{x})_i = \mathbf{x}'\}$ as the "reverse neighborhood" set, where an element $(\mathbf{x}, i) \in \mathcal{N}^{-1}(\mathbf{x}')$ indicates that $\mathbf{x}'$ is the $i$-th neighbor of $\mathbf{x}$. There could be multiple $\mathbf{x}$ with the same $i$ in $\mathcal{N}^{-1}(\mathbf{x}')$ as in the case of the star graph. Computing and storing $\mathcal{N}^{-1}$ as a hash table mapping $\mathbf{x}$ to the set of $\mathcal{N}^{-1}(\mathbf{x})$ has a time and space complexity of at most $O(\sum_{\mathbf{x}} |\mathcal{N}(\mathbf{x})|)$ (*i.e.*, the number of edges). For special structures (*e.g.*, grid), we can design specific algorithms that bypass this process. We provide more details in Appendix C.

---

**Algorithm 1** An unbiased estimator of $\mathcal{J}_1$
**Input:** $p_{\text{data}}, \mathcal{N}, \boldsymbol{c}_\theta$ (model).
1. Sample $\mathbf{x} \sim p_{\text{data}}(\mathbf{x})$.
2. Sample $i \sim \text{Uniform}\{1, \ldots, |\mathcal{N}(\mathbf{x})|\}$.
**Output:** $|\mathcal{N}(\mathbf{x})| \cdot \big(\boldsymbol{c}_\theta(\mathbf{x}; \mathcal{N})_i^2 + 2\boldsymbol{c}_\theta(\mathbf{x}; \mathcal{N})_i\big)$.

**Algorithm 2** An unbiased estimator of $\mathcal{J}_2$
**Input:** $p_{\text{data}}, \mathcal{N}, \boldsymbol{c}_\theta$ (model).
1. Sample $\mathbf{x}' \sim p_{\text{data}}(\mathbf{x})$.
2. Sample $(\mathbf{x}, i) \sim \text{Uniform}(\mathcal{N}^{-1}(\mathbf{x}'))$.
**Output:** $2 \cdot |\mathcal{N}^{-1}(\mathbf{x}')| \cdot \boldsymbol{c}_\theta(\mathbf{x}; \mathcal{N})_i$.

---

We leverage the grid structure in our experiments, where the neighbors of $\mathbf{x}$ are defined to be $\mathcal{N}(\mathbf{x}) = \{\mathbf{x} \pm \mathbf{e}_d\}_{d=1}^D$. Algorithm 1 and Algorithm 2 allow us to efficiently train the model by sampling a dimension $d \in [D]$ and flipping the bit for the $d$th entry of $\mathbf{x}$ for each $\mathbf{x}_n$.

## 4.3 Denoising Concrete Score Matching

Finally, we propose another training objective to approximate Eq. 5, with a focus on computational efficiency. Similar to denoising score matching (DSM) for (Stein) score estimation [22], we derive a denoising counterpart for Concrete score estimation called "Denoising Concrete Score Matching" (D-CSM). Specifically, given a discrete data distribution $p_{\text{data}}(\mathbf{x})$ and a discrete noise distribution $\tilde{q}(\tilde{\mathbf{x}}|\mathbf{x})$, we define the perturbed data distribution $\tilde{p}(\tilde{\mathbf{x}}) = \sum_{\mathbf{x}} p_{\text{data}}(\mathbf{x})\tilde{q}(\tilde{\mathbf{x}}|\mathbf{x})$, and the posterior $q(\mathbf{x}|\tilde{\mathbf{x}}) = \frac{p_{\text{data}}(\mathbf{x})\tilde{q}(\tilde{\mathbf{x}}|\mathbf{x})}{\tilde{p}(\tilde{\mathbf{x}})}$. Then, we can show that the Concrete score of the perturbed data distribution $\tilde{p}(\tilde{\mathbf{x}})$ can be obtained via $\boldsymbol{c}_{\tilde{p}(\tilde{\mathbf{x}})}(\tilde{\mathbf{x}}; \mathcal{N}) = \sum_{\mathbf{x}} \boldsymbol{c}_{\tilde{q}(\tilde{\mathbf{x}}|\mathbf{x})}(\tilde{\mathbf{x}}; \mathcal{N})q(\mathbf{x}|\tilde{\mathbf{x}})$. This property allows us to obtain the D-CSM objective, as shown in the following theorem:

**Theorem 4** (Denoising Concrete Score Matching). *The objective:*

$$\mathcal{J}_{\text{D}-\text{CSM}}(\theta) = \sum_{\mathbf{x}, \tilde{\mathbf{x}}} p_{\text{data}}(\mathbf{x})\tilde{q}(\tilde{\mathbf{x}}|\mathbf{x}) \left\| \boldsymbol{c}_\theta(\tilde{\mathbf{x}}; \mathcal{N}) - \boldsymbol{c}_{q(\tilde{\mathbf{x}}|\mathbf{x})}(\tilde{\mathbf{x}}; \mathcal{N}) \right\|_2^2 \tag{6}$$

*is minimized when* $\boldsymbol{c}_\theta(\tilde{\mathbf{x}}; \mathcal{N}) = \boldsymbol{c}_{p(\tilde{\mathbf{x}})}(\tilde{\mathbf{x}}; \mathcal{N})$.

We draw a direct analogy to DSM in order to provide additional insight into D-CSM. Recall that in DSM, there exists a noise distribution $q(\tilde{\mathbf{x}}|\mathbf{x})$ (*i.e.* $\mathcal{N}(\tilde{\mathbf{x}}|\mathbf{x}, \sigma^2 I)$) such that the magnitude of Stein scores captures the distance between the perturbed $\tilde{\mathbf{x}}$ and its clean counterpart $\mathbf{x}$. D-CSM enjoys a similar interpretation. In particular, consider an input space $\mathbf{x} \in \mathcal{X} \subseteq \mathbb{Z}^D$ with the neighborhood structure $\mathcal{N}(\mathbf{x}) = \{\mathbf{x} + \mathbf{v}|\mathbf{v} \in \{-1, 0, 1\}^D\}$. If we define $q(\tilde{\mathbf{x}}|\mathbf{x}) = \frac{1}{2^{D+k}}$ where $k$ is the Manhattan distance between $\tilde{\mathbf{x}}$ and $\mathbf{x}$, then the Concrete score $\boldsymbol{c}_{\tilde{p}(\tilde{\mathbf{x}})}(\tilde{\mathbf{x}}; \mathcal{N})$ captures how close $\mathbf{x}$ is to $\tilde{\mathbf{x}}$ as measured by the Manhattan distance. Therefore, D-CSM can also be understood through the lens of a denoiser. We provide additional experimental results exploring the performance of D-CSM in Appendix B.3.

## 5 Experimental Results

In this section, we evaluate the performance of CSM on synthetic, tabular, and discrete image datasets on a variety of sampling and density estimation tasks. We provide additional details on specific experimental settings and hyperparameter configurations in Appendix C.

## 5.1 Baselines

In our experiments, we compare CSM to two relevant baselines for modeling discrete data: Ratio Matching and Discrete Score Matching with Marginalization (`Discrete Marginalization`). For conciseness, we provide additional details and derivations for their training objectives in Appendix D.

**Ratio Matching.** Similar to score matching, Ratio Matching [34] leverages the fact that ratios of probabilities are independent of the intractable normalizing constant (due to cancellation). The method then seeks to match the ground truth density ratios $\frac{p_{\text{data}}(\mathbf{x})}{p_{\text{data}}(\mathbf{x}_{-d})} = \frac{q_\theta(\mathbf{x})}{q_\theta(\mathbf{x}_{-d})}$ where $\mathbf{x}_{-d}$ denotes the vector $\mathbf{x}$ with the $d$th entry bit-flipped (e.g. from 0 to 1).

**Discrete Score Matching with Marginalization (Discrete Marginalization).** The Discrete Marginalization baseline is another way of estimating discrete probability distributions with score matching as in [34]. However, we note that this approach is difficult to scale because it requires us to marginalize over the data dimension, which may also cause instabilities during training.

## 5.2 1-D Discrete Data

In this experiment, we consider a 16-category 1-D data distribution as shown in Figure 2. We parameterize our Concrete score model $\boldsymbol{c}_\theta(\mathbf{x}, \mathcal{N})$ using a shallow MLP model with Tanh activations, and use the Cycle neighborhood structure during training. We demonstrate that the learned Concrete score model can faithfully capture the true data distribution when trained with CSM. As shown on the left side of Figure 2, we find that CSM generates samples (blue) using MH that almost perfectly matches the samples drawn from $p_{\text{data}}(\mathbf{x})$ (green).

We also observe in Appendix A.3 that the Concrete score on $p_{\text{data}}(\mathbf{x})$ can recover the Stein score of a data distribution perturbed with triangular noise. Even though the Concrete score model is trained on discrete data (green), this connection allows us to sample with Langevin dynamics using the recovered Stein score. On the right side of Figure 2, we show how the samples generated using Langevin dynamics (smoothed orange) closely match the triangular noise-perturbed data distribution (gray). Then, we leverage a closed-form denoising formula for the perturbed distribution (Appendix A.3) to recover the clean data distribution (green) from samples obtained via Langevin dynamics (smoothed orange). The resulting denoised samples (orange) are labeled as `Denoised` in Figure 2. We provide more details and discussion in Appendix A.3.

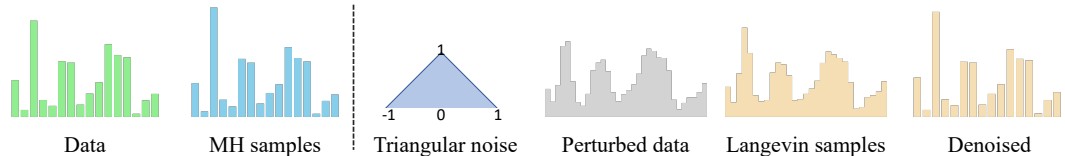

| Data | MH samples | Triangular noise | Perturbed data | Langevin samples | Denoised |

Figure 2: Sampling results from a toy 1-D discrete dataset. CSM recovers the true data distribution $p_{\text{data}}(\mathbf{x})$ using Metropolis-Hastings (left). The Concrete score can also be used to recover the Stein score of the triangular noise-perturbed data distribution, allowing for sampling with Langevin dynamics. Such samples can be denoised to recover the original (clean) data distribution (right).

## 5.3 Sampling with Toy 2-D Multi-Class Datasets

Next, we consider three 2-D toy benchmark datasets with multiple modes and discontinuities as commonly used in the density estimation literature [40, 41]. We quantize the data into $91 \times 91$ bins to obtain the discrete training data for the experiment (see Figure 3). We compare our method against the two baselines: (1) Ratio Matching [42]; and (2) Discrete Marginalization [34]. In particular, we found that the original equations in [34] were incorrect, and provide results using the correct training objectives (denoted as `Ratio-fixed` and `Marginal-fixed`) in Figure 3. We provide a step-by-step derivation addressing this issue with the correct expressions for the training objectives in Appendix D.

For a fair comparison, we use the same model architecture and training configurations across all methods. We use the grid neighborhood structure for training CSM. As shown in Figure 3, the samples from CSM best capture the shape of the underlying data distributions across all three datasets (leftmost column). Baselines implemented using the original equations in [34] indeed demonstrate poor performance on the density estimation task (`Ratio` and `Marginal` columns) in Figure 3.

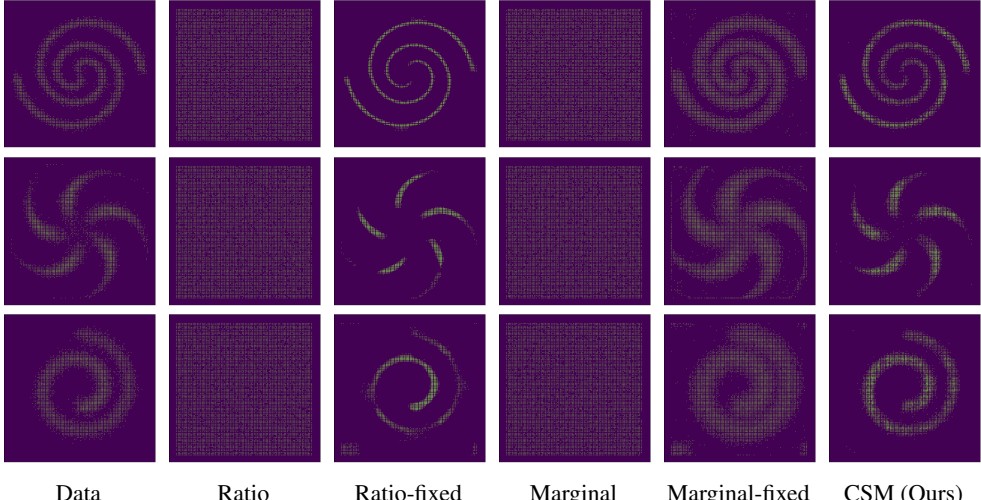

| Data | Ratio | Ratio-fixed | Marginal | Marginal-fixed | CSM (Ours) |

Figure 3: Sampling results on toy 2-D benchmark datasets. We find that CSM produces the highest quality samples across all 3 datasets relative to all baselines.

| Datasets | Ratio Matching ($\uparrow$) | Discrete Marginalization ($\uparrow$) | CSM (Ours) ($\uparrow$) |
|---|---|---|---|
| NLTCS | -6.15 | -6.21 | **-6.13** |
| Plants | -15.44 | -19.03 | **-14.02** |
| Jester | -56.49 | -57.06 | **-54.91** |
| Amazon Diaper | **-10.69** | -42.52 | -11.13 |
| Amazon Feeding | **-12.09** | -35.96 | -12.65 |
| Amazon Gifts | -4.57 | -4.28 | **-4.22** |
| Amazon Media | **-10.22** | -13.77 | -10.30 |
| Amazon Toys | -9.83 | -16.34 | **-9.30** |

Table 1: Log-likelihood comparisons on discrete tabular datasets. Higher is better. We find that CSM demonstrates good performance, almost always outperforming or performing comparably relative to the `Ratio Matching` and `Discrete Marginalization` baselines.

## 5.4 Likelihood Evaluation on Discrete Tabular Data

We then evaluate the performance of CSM on density estimation tasks for a wide range of tabular (discrete) datasets drawn from both the Twenty Datasets [43] and the Amazon Baby Registries benchmarks [44]. In order to evaluate likelihoods, we directly parameterize the density with a discrete autoregressive model (MADE [45]), but train the model using the baseline approaches and CSM. For a fair comparison, we use the same model architecture and experimental configurations across all methods. Similar to our previous experiments, we use the grid neighborhood structure for training CSM. As shown in Table 1, our approach (`CSM`) demonstrates favorable performance relative to the `Ratio Matching` and `Discrete Marginalization` baselines. We provide additional experimental results in Appendix B.1 and more experimental details in Appendix C.

## 5.5 Generative Modeling with High-dimensional Images

For our final experiment, we demonstrate that we can scale CSM to achieve good performance on complex, high-dimensional binarized image datsets. We experiment with the MNIST [46] dataset, which has 784 dimensions. We directly parameterize the Concrete score function with a U-Net [47], and train the model using the CSM based on a grid neighborhood structure. Similar to [26], we perturb the image with different levels of discrete (Categorical) noise, and train the models at different noise levels with annealing. After the Concrete score model is trained, we sample from the model using Metropolis-Hastings as discussed in Section 3.3 and follow a similar sample initialization process as [26]. We provide more details in Appendix C. We present the samples from our model in Figure 4b. We observe that our CSM approach with Metropolis-Hastings is able to generate samples that look similar to the digit examples in the training set (Figure 4a).

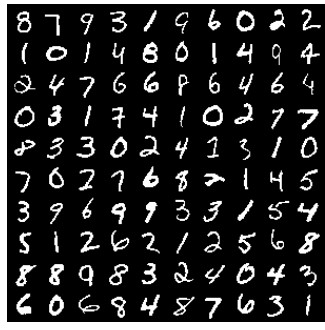
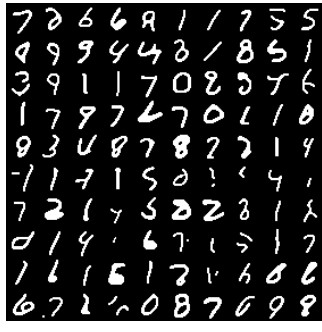

| (a) Binary training images | (b) MNIST samples (MH) |

Figure 4: Image samples from CSM on the binarized MNIST dataset. We observe that CSM is able generate high-quality samples on MNIST using Metropolis-Hastings.

# 6   Related Work

**Score Matching.** Our work builds on the body of literature on score matching [21, 22] and score-based generative modeling [23, 26, 16]. Notably, we build upon the generalization of score matching to discrete data [42, 48, 34]. Our score function can be viewed through the lens of generalized score matching as in [34], where the Concrete score serves as a particular instantiation of the linear operator in their framework. However, we provide a novel perspective in terms of constructing the Concrete score via surrogate gradient information over a structured discrete space, and provide efficient training objectives for CSM. Our work is also closely related to score matching with finite difference (FD-SM) [33] as discussed in Section 3.2. However, rather than approximating the gradient with finite differences for continuous data, we leverage the first-order forward difference to construct our Concrete score function in discrete domains. In addition, CSM bears similarities to a concurrent work on scaling up Ratio Matching [34] for discrete energy-based models (EBMs) [49] called RMwGGIS— our approach can be viewed as a different way to train discrete EBMs (via the Concrete score).

**Generative Modeling for Discrete Data.** CSM also provides another way to train generative models for discrete data. A related work is [50], which trains a score-based model on distributions over graphs. However, they perturb the adjacency matrices with Gaussian noise and use the continuous variant of DSM. Although there exist several model families for learning binary and Categorical probability distributions such as normalizing flows [51–53], Sum-Product Networks (SPNs) [54, 55], denoising diffusion probabilistic models [56], discrete (latent) [57, 58] EBMs [49], and Generative Flow Networks (GFNs) [59], there does not yet exist a discrete score-based model that can scale to high-dimensional discrete datasets. Finally, our approach bears similarities with Gibbs with Gradients (GWG) [60]. In particular, GWG augments existing MCMC samplers with gradient information by leveraging local structure to estimate likelihood ratios for transitioning to the next state. However, GWG utilizes gradients from the probability mass functions of the underlying discrete distributions, in contrast to the way in which we construct the Concrete score function.

# 7   Conclusion

We introduced Concrete Score Matching (CSM), a novel framework for learning discrete probability distributions via score matching. We proposed to leverage particular kinds of structural information in the data to construct *surrogate gradient information* about the discrete space, and used this to define a valid score function. We also introduced several modifications to the original training objective that allowed CSM to scale gracefully to high-dimensional datasets, and demonstrated that CSM performs well on a variety of sampling and density estimation tasks.

However, this work is not without limitations. Since CSM depends on the neighborhood structure, certain types of graphs may work better for score matching in practice than others. Additionally, our efficient training objectives may suffer from high variance when scaling to high dimensions for particular kinds of neighborhood-induced graphs, such as the Star graph. For future work, although we fixed the number of neighbors $K$ for each $\mathbf{x}$ in our experiments, it would be interesting to adaptively determine the optimal number of neighbors to use for each $\mathbf{x}$ during training. We also

believe that CSM can be generalized to leverage more complex neighborhood structures than those we explored in the paper. Additionally, empirically investigating the performance of D-CSM with Langevin dynamics relative to CSM would be interesting future work.

**Broader Impact.** This work introduces a novel score function—the Concrete score—that is defined over discrete spaces, as well as a corresponding score matching (CSM) framework that can scale to high-dimensional discrete datasets. This leads to empirical performance improvements over a range of density estimation and sampling tasks, and does not have a direct consequence on societal issues. However, we note that CSM could serve as the basis for developing more powerful generative models of structured, discrete data. Although this could lead to tangible benefits (e.g. improved generative modeling of text data), we should be mindful to take the usual precautions required for generative modeling research (e.g. against the development of deepfakes).

## Acknowledgements and Funding Disclosure

We thank the anonymous reviewers for insightful discussions and feedback. KC is supported by the Qualcomm Innovation Fellowship and the Two Sigma Diversity PhD Fellowship. This research was supported by NSF (#1651565), AFOSR (FA95501910024), ARO (W911NF-21-1-0125), ONR, DOE, CZ Biohub, and Sloan Fellowship.

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
