# Appendix

## A  Proofs of Theoretical Results

In this section, we provide proofs and additional discussions of the theoretical results.

### A.1  Concrete Score Matching

**Theorem 1** (Completeness). *Let $p_{\mathrm{data}}(\mathbf{x})$ be a (discrete) data distribution. Denote $\boldsymbol{c}_{p_{\mathrm{data}}}(\mathbf{x}; \mathcal{N})$ as the Concrete score of $p_{\mathrm{data}}(\mathbf{x})$ with neighboring structure $\mathcal{N}$, and $\boldsymbol{c}_{\theta}(\mathbf{x}; \mathcal{N})$ the Concrete score for a distribution $p_{\theta}(\mathbf{x})$ parameterized by $\theta \in \Theta$. When the graph induced by the neighborhood structure $\mathcal{N}$ is connected, $\boldsymbol{c}_{\theta}(\mathbf{x}; \mathcal{N}) = \boldsymbol{c}_{p_{\mathrm{data}}}(\mathbf{x}; \mathcal{N})$ implies that $p_{\theta}(\mathbf{x}) = p_{\mathrm{data}}(\mathbf{x}) \ \forall \mathbf{x} \in \mathcal{X}$.*

*Proof.* If two nodes $\mathbf{x}$ and $\mathbf{x}'$ in a neighborhood-induced graph $\mathcal{G}$ share an edge, then their density ratio $p_{\mathrm{data}}(\mathbf{x}')/p_{\mathrm{data}}(\mathbf{x})$ and $p_{\mathrm{data}}(\mathbf{x})/p_{\mathrm{data}}(\mathbf{x}')$ can be uniquely identified using $\boldsymbol{c}_{p_{\mathrm{data}}}(\mathbf{x}; \mathcal{N})$ based on the definition:

$$\boldsymbol{c}_{p_{\mathrm{data}}}(\mathbf{x}; \mathcal{N}) + \mathbf{1} = \left[ \frac{p_{\mathrm{data}}(\mathbf{x}_{n_1})}{p_{\mathrm{data}}(\mathbf{x})}, ..., \frac{p_{\mathrm{data}}(\mathbf{x}_{n_k})}{p_{\mathrm{data}}(\mathbf{x})} \right]^T . \tag{7}$$

For simplicity, we name the elements in $\mathcal{X}$: $\{\mathbf{x}_1, \mathbf{x}_2, ..., \mathbf{x}_N\}$. Note that when $\mathcal{G}$ is a weakly connected graph, any node pair $(\mathbf{x}_1, \mathbf{x}_n)$ is connected via a path in the graph. Denoting the path between these two nodes as $\mathbf{x}_1 \to ... \to \mathbf{x}_n$, we can obtain the density ratio for any neighboring pairs on the path using the definition of $\boldsymbol{c}_{p_{\mathrm{data}}}(\mathbf{x}; \mathcal{N})$. Therefore the density ratio $p_{\mathrm{data}}(\mathbf{x}_n)/p_{\mathrm{data}}(\mathbf{x}_1)$ can be computed using the products of the density ratios for all neighboring data pairs on the path. This implies that when $\mathcal{G}$ is weakly connected, we can uniquely recover the density ratios $\{p_{\mathrm{data}}(\mathbf{x}_1)/p_{\mathrm{data}}(\mathbf{x}_1), p_{\mathrm{data}}(\mathbf{x}_2)/p_{\mathrm{data}}(\mathbf{x}_1), \ldots, p_{\mathrm{data}}(\mathbf{x}_N)/p_{\mathrm{data}}(\mathbf{x}_1)\}$, which uniquely determine $p_{\mathrm{data}}(\mathbf{x})$. Therefore, knowing the density ratio between any two points uniquely identifies $p_{\mathrm{data}}(\mathbf{x})$. $\square$

**Theorem 2** (Consistency). *In the limit of infinite data and infinite model capacity, the optimal $\theta^*$ that minimizes Eq. 4 recovers the true Concrete score, or satisfies $\boldsymbol{c}_{\theta^*}(\mathbf{x}; \mathcal{N}) = \boldsymbol{c}_{p_{\mathrm{data}}}(\mathbf{x}; \mathcal{N})$.*

*Proof.* It is easy to see the optimal $\theta^*$ that minimizes the following equation

$$\mathcal{L}_{\mathrm{CSM}}(\theta) = \sum_{\mathbf{x}} p_{\mathrm{data}}(\mathbf{x}) \| \boldsymbol{c}_{\theta}(\mathbf{x}; \mathcal{N}) - \boldsymbol{c}_{p_{\mathrm{data}}}(\mathbf{x}; \mathcal{N}) \|_2^2 \tag{8}$$

satisfies $\boldsymbol{c}_{\theta^*}(\mathbf{x}; \mathcal{N}) = \boldsymbol{c}_{p_{\mathrm{data}}}(\mathbf{x}; \mathcal{N})$ almost everywhere. $\square$

**Theorem 3.** *Optimizing Equation (4) is equivalent to optimizing:*

$$\mathcal{J}_{\mathrm{CSM}}(\theta) = \underbrace{\sum_{\mathbf{x}} \sum_{i=1}^{|\mathcal{N}(\mathbf{x})|} p_{\mathrm{data}}(\mathbf{x}) \left( \boldsymbol{c}_{\theta}(\mathbf{x}; \mathcal{N})_i^2 + 2 \boldsymbol{c}_{\theta}(\mathbf{x}; \mathcal{N})_i \right)}_{\mathcal{J}_1} - \underbrace{\sum_{\mathbf{x}} \sum_{i=1}^{|\mathcal{N}(\mathbf{x})|} 2 p_{\mathrm{data}}(\mathbf{x}_{n_i}) \boldsymbol{c}_{\theta}(\mathbf{x}; \mathcal{N})_i}_{\mathcal{J}_2} \tag{5}$$

*where $\mathcal{N}(\mathbf{x}) = \{\mathbf{x}_{n_1}, ..., \mathbf{x}_{n_k}\}$ is the set of neighbors of $\mathbf{x}$.*

*Proof.* Recall that by definition

$$\boldsymbol{c}_{p_{\mathrm{data}}}(\mathbf{x}; \mathcal{N}) \triangleq \left[ \frac{p_{\mathrm{data}}(\mathbf{x}_{n_1}) - p_{\mathrm{data}}(\mathbf{x})}{p_{\mathrm{data}}(\mathbf{x})}, ..., \frac{p_{\mathrm{data}}(\mathbf{x}_{n_k}) - p_{\mathrm{data}}(\mathbf{x})}{p_{\mathrm{data}}(\mathbf{x})} \right]^T . \tag{9}$$

$$\arg\min_\theta \mathcal{L}_{\text{CSM}}(\theta)$$

$$= \arg\min_\theta \sum_{\mathbf{x}} p_{\text{data}}(\mathbf{x}) \left\| \mathbf{c}_\theta(\mathbf{x};\mathcal{N}) - \mathbf{c}_{p_{\text{data}}}(\mathbf{x};\mathcal{N}) \right\|_2^2$$

$$= \arg\min_\theta \sum_{\mathbf{x}} p_{\text{data}}(\mathbf{x}) \left[ \left\| \mathbf{c}_{p_{\text{data}}}(\mathbf{x};\mathcal{N}) \right\|_2^2 - 2\mathbf{c}_\theta(\mathbf{x};\mathcal{N})^T \mathbf{c}_{p_{\text{data}}}(\mathbf{x};\mathcal{N}) + \left\| \mathbf{c}_\theta(\mathbf{x};\mathcal{N}) \right\|_2^2 \right]$$

$$= \arg\min_\theta \sum_{\mathbf{x}} p_{\text{data}}(\mathbf{x}) \left[ \left\| \mathbf{c}_\theta(\mathbf{x};\mathcal{N}) \right\|_2^2 - 2\mathbf{c}_\theta(\mathbf{x};\mathcal{N})^T \mathbf{c}_{p_{\text{data}}}(\mathbf{x};\mathcal{N}) \right]$$

$$= \arg\min_\theta \underbrace{\sum_{\mathbf{x}} \sum_{i=1}^{|\mathcal{N}(\mathbf{x})|} p_{\text{data}}(\mathbf{x}) \left( \mathbf{c}_\theta(\mathbf{x};\mathcal{N})_i^2 + 2\mathbf{c}_\theta(\mathbf{x};\mathcal{N})_i \right)}_{\mathcal{J}_1} - \underbrace{\sum_{\mathbf{x}} \sum_{i=1}^{|\mathcal{N}(\mathbf{x})|} 2p_{\text{data}}(\mathbf{x}_{n_i}) \mathbf{c}_\theta(\mathbf{x};\mathcal{N})_i}_{\mathcal{J}_2}$$

$$= \arg\min_\theta \mathcal{J}_{\text{CSM}}(\theta)$$

$\square$

## A.2 Denoising Concrete Score Matching

**Property 1.** $\mathbf{c}_{\tilde{p}(\tilde{\mathbf{x}})}(\tilde{\mathbf{x}};\mathcal{N}) = \sum_{\mathbf{x}} \mathbf{c}_{\tilde{q}(\tilde{\mathbf{x}}|\mathbf{x})}(\tilde{\mathbf{x}};\mathcal{N}) q(\mathbf{x}|\tilde{\mathbf{x}})$

*Proof.* For simplicity, given a distribution $p(\mathbf{x})$, we define the operator $L[p(\mathbf{x})] = \left[ p(\mathbf{x}_{n_1}) - p(\mathbf{x}), ..., p(\mathbf{x}_{n_k}) - p(\mathbf{x}) \right]^T$, where $\mathcal{N}(\mathbf{x}) = \{\mathbf{x}_{n_1}, \ldots, \mathbf{x}_{n_k}\}$ are the neighbors of $\mathbf{x}$. Recall that by definition, $\tilde{p}(\tilde{\mathbf{x}}) = \sum_{\mathbf{x}} p_{\text{data}}(\mathbf{x})\tilde{q}(\tilde{\mathbf{x}}|\mathbf{x})$ and the posterior $q(\mathbf{x}|\tilde{\mathbf{x}}) = \frac{p_{\text{data}}(\mathbf{x})\tilde{q}(\tilde{\mathbf{x}}|\mathbf{x})}{\tilde{p}(\tilde{\mathbf{x}})}$. We have

$$\begin{aligned}
c_{\tilde{p}(\tilde{\mathbf{x}})}(\tilde{\mathbf{x}};\mathcal{N}) &= \frac{L[\tilde{p}(\tilde{\mathbf{x}})]}{\tilde{p}(\tilde{\mathbf{x}})} \\
&= \frac{L[\sum_{\mathbf{x}} p_{\text{data}}(\mathbf{x})\tilde{q}(\tilde{\mathbf{x}}|\mathbf{x})]}{\tilde{p}(\tilde{\mathbf{x}})} \\
&= \frac{\sum_{\mathbf{x}} p_{\text{data}}(\mathbf{x}) L[\tilde{q}(\tilde{\mathbf{x}}|\mathbf{x})]}{\tilde{p}(\tilde{\mathbf{x}})} \quad \text{(by linearity)} \\
&= \sum_{\mathbf{x}} \frac{L[\tilde{q}(\tilde{\mathbf{x}}|\mathbf{x})]}{\tilde{q}(\tilde{\mathbf{x}}|\mathbf{x})} \frac{p_{\text{data}}(\mathbf{x})\tilde{q}(\tilde{\mathbf{x}}|\mathbf{x})}{\tilde{p}(\tilde{\mathbf{x}})} \\
&= \sum_{\mathbf{x}} c_{\tilde{q}(\tilde{\mathbf{x}}|\mathbf{x})}(\tilde{\mathbf{x}};\mathcal{N}) q(\mathbf{x}|\tilde{\mathbf{x}})
\end{aligned}$$

$\square$

**Theorem 4** (Denoising Concrete Score Matching). *The objective:*

$$\mathcal{J}_{\text{D-CSM}}(\theta) = \sum_{\mathbf{x},\tilde{\mathbf{x}}} p_{\text{data}}(\mathbf{x})\tilde{q}(\tilde{\mathbf{x}}|\mathbf{x}) \left\| \mathbf{c}_\theta(\tilde{\mathbf{x}};\mathcal{N}) - \mathbf{c}_{q(\tilde{\mathbf{x}}|\mathbf{x})}(\tilde{\mathbf{x}};\mathcal{N}) \right\|_2^2 \tag{6}$$

*is minimized when* $\mathbf{c}_\theta(\tilde{\mathbf{x}};\mathcal{N}) = \mathbf{c}_{p(\tilde{\mathbf{x}})}(\tilde{\mathbf{x}};\mathcal{N})$.

*Proof.* The model $\mathbf{c}_\theta$ that minimizes the least squares

$$\theta^* = \arg\min_\theta \sum_{\mathbf{x},\tilde{\mathbf{x}}} p_{\text{data}}(\mathbf{x})\tilde{q}(\tilde{\mathbf{x}}|\mathbf{x}) \| \mathbf{c}_\theta(\tilde{\mathbf{x}};\mathcal{N}) - c_{\tilde{q}(\mathbf{x}|\tilde{\mathbf{x}})}(\tilde{\mathbf{x}};\mathcal{N}) \|^2 \tag{10}$$

satisfies $\mathbf{c}_\theta(\tilde{\mathbf{x}};\mathcal{N}) = \sum_{\mathbf{x}} c_{\tilde{q}(\tilde{\mathbf{x}}|\mathbf{x})}(\tilde{\mathbf{x}};\mathcal{N}) q(\mathbf{x}|\tilde{\mathbf{x}}) = c_{\tilde{p}(\tilde{\mathbf{x}})}(\tilde{\mathbf{x}};\mathcal{N})$ using Property 1. $\square$

## A.3 Connection to distribution perturbed with triangular noise

As discussed in Section 5.2, our approach can be used to denoise a given data distribution perturbed with triangular noise. For simplicity, we assume $\mathbf{x} \in \mathcal{X} \subseteq \mathbb{Z}^D$. The argument proceeds in a similar manner to how denoising score matching can be used to compute the expected posterior $\mathbb{E}_{\mathbf{x}}[p(\mathbf{x}|\tilde{\mathbf{x}})]$, where $\tilde{\mathbf{x}}$ is the perturbed data.

**Definition 3** (Triangular noise). *We define the PDF of the $D$-dimensional triangular distribution with lower limit -1, upper limit 1, and mode 0 as the following:*

$$\mathcal{T}_D(\mathbf{x}) = \begin{cases} 0 & \mathbf{x} < -1 \\ \mathbf{x} + 1 & -1 \leq \mathbf{x} \leq 0 \\ 1 - \mathbf{x} & 0 < \mathbf{x} \leq 1 \\ 0 & \mathbf{x} > 1 \end{cases} \tag{11}$$

**Lemma 1.** *Given a $D$-dimensional discrete distribution $p_{\text{data}}(\mathbf{x})$, let $\tilde{p}(\tilde{\mathbf{x}})$ be the distribution of $p_{\text{data}}(\mathbf{x})$ when perturbed with triangular noise as defined in Equation* (11).

$$\tilde{p}(\tilde{\mathbf{x}}) \triangleq \sum_{\mathbf{x}} p_{\text{data}}(\mathbf{x})\mathcal{T}_D(\tilde{\mathbf{x}} - \mathbf{x}). \tag{12}$$

*Then for any $\mathbf{x}, \mathbf{y} \sim p_{\text{data}}(\mathbf{x})$ we have $\frac{\tilde{p}(\mathbf{x})}{\tilde{p}(\mathbf{y})} = \frac{p_{\text{data}}(\mathbf{x})}{p_{\text{data}}(\mathbf{y})}$.*

*Proof.* It is easy to see that for any $\mathbf{x}, \mathbf{y} \in \mathcal{X} \subseteq \mathbb{Z}^D$, we have

$$\tilde{p}(\mathbf{x}) = p_{\text{data}}(\mathbf{x})\mathcal{T}_D(\mathbf{0}), \ \tilde{p}(\mathbf{y}) = p_{\text{data}}(\mathbf{y})\mathcal{T}_D(\mathbf{0}) \tag{13}$$

because the width of the triangular noise is less than one. Thus $\frac{\tilde{p}(\mathbf{x})}{\tilde{p}(\mathbf{y})} = \frac{p_{\text{data}}(\mathbf{x})}{p_{\text{data}}(\mathbf{y})}$.

$\square$

Denote $\bar{\mathbf{x}}_d$ the integer such that the $d$-th index of $\tilde{\mathbf{x}}$, denoted $\tilde{\mathbf{x}}_d$, satisfies $\tilde{\mathbf{x}}_d \in [\bar{\mathbf{x}}_d, \bar{\mathbf{x}}_d + 1)$. Similar to Section 4.3, in the discrete case with perturbed triangular noise $\mathcal{T}_D(\mathbf{x})$, we define $\tilde{p}(\tilde{\mathbf{x}}) = \sum_{\mathbf{x}} p_{\text{data}}(\mathbf{x})\mathcal{T}_D(\tilde{\mathbf{x}}|\mathbf{x})$ and the posterior $q(\mathbf{x}|\tilde{\mathbf{x}}) = \frac{p_{\text{data}}(\mathbf{x})\mathcal{T}_D(\tilde{\mathbf{x}}|\mathbf{x})}{\tilde{p}(\tilde{\mathbf{x}})}$, where $\mathcal{T}_D(\tilde{\mathbf{x}}|\mathbf{x}) = \mathcal{T}_D(\tilde{\mathbf{x}} - \mathbf{x})$.

We have

$$\mathbb{E}_{\mathbf{x}}[q(\mathbf{x}|\tilde{\mathbf{x}})] = \sum_{\mathbf{x} \in \{\mathbf{x}_d \in \{\bar{\mathbf{x}}_d, \bar{\mathbf{x}}_d+1\}\}} \mathbf{x}q(\mathbf{x}|\tilde{\mathbf{x}}) \tag{14}$$

$$= \sum_{\mathbf{x} \in \{\mathbf{x}_d \in \{\bar{\mathbf{x}}_d, \bar{\mathbf{x}}_d+1\}\}} \mathbf{x}\frac{p_{\text{data}}(\mathbf{x})\mathcal{T}_D(\tilde{\mathbf{x}}|\mathbf{x})}{p(\tilde{\mathbf{x}})} \tag{15}$$

Using the fact that $\mathcal{T}_D$ is independent among dimensions, we have

$$p_{\text{data}}(\mathbf{x})\mathcal{T}_D(\tilde{\mathbf{x}}|\mathbf{x}) = p_{\text{data}}(\mathbf{x}) \prod_{d=1}^{D} \left( \mathbb{1}[\mathbf{x}_d = \bar{\mathbf{x}}_d] * (\bar{\mathbf{x}}_d + 1 - \tilde{\mathbf{x}}_d) + \mathbb{1}[\mathbf{x}_d = \bar{\mathbf{x}}_d + 1] * (\tilde{\mathbf{x}}_d - \bar{\mathbf{x}}_d) \right) \tag{16}$$

and

$$\tilde{p}(\tilde{\mathbf{x}}) = \sum_{\mathbf{x} \in \{\mathbf{x}_d \in \{\bar{\mathbf{x}}_d, \bar{\mathbf{x}}_d+1\}\}} p_{\text{data}}(\mathbf{x}) \prod_{d=1}^{D} \left( \mathbb{1}[\mathbf{x}_d = \bar{\mathbf{x}}_d] * (\bar{\mathbf{x}}_d + 1 - \tilde{\mathbf{x}}_d) + \mathbb{1}[\mathbf{x}_d = \bar{\mathbf{x}}_d + 1] * (\tilde{\mathbf{x}}_d - \bar{\mathbf{x}}_d) \right) \tag{17}$$

Plugging in Equation (16) and Equation (17), we have

$$q(\mathbf{x}|\tilde{\mathbf{x}}) = \frac{\prod_{d=1}^{D} \left( \mathbb{1}[\mathbf{x}_d = \bar{\mathbf{x}}_d] * (\bar{\mathbf{x}}_d + 1 - \tilde{\mathbf{x}}_d) + \mathbb{1}[\mathbf{x}_d = \bar{\mathbf{x}}_d + 1] * (\tilde{\mathbf{x}}_d - \bar{\mathbf{x}}_d) \right)}{\sum_{\mathbf{y} \in \{\mathbf{y}_d \in \{\bar{\mathbf{x}}_d, \bar{\mathbf{x}}_d+1\}\}} \frac{p_{\text{data}}(\mathbf{y})}{p_{\text{data}}(\mathbf{x})} \prod_{d=1}^{D} \left( \mathbb{1}[\mathbf{y}_d = \bar{\mathbf{x}}_d] * (\bar{\mathbf{x}}_d + 1 - \tilde{\mathbf{x}}_d) + \mathbb{1}[\mathbf{y}_d = \bar{\mathbf{x}}_d + 1] * (\tilde{\mathbf{x}}_d - \bar{\mathbf{x}}_d) \right)}. \tag{18}$$

Equation (18) means that given $\tilde{\mathbf{x}}$, we can compute the posterior in closed form using density ratios $\frac{p_{\text{data}}(\mathbf{y})}{p_{\text{data}}(\mathbf{x})}$, which can be evaluated using Concrete scores as long as the graph induced by the neighborhood structure is connected (based on Theorem 1). Knowing $p(\mathbf{x}|\tilde{\mathbf{x}})$ also allows us to sample from $p(\mathbf{x}|\tilde{\mathbf{x}})$ to perform denoising. Similarly, given $q(\mathbf{x}|\tilde{\mathbf{x}})$, we can also compute Equation (14) in closed form.

**Recovering Stein Scores** Given Concrete scores, we can uniquely recover the Stein score of $\tilde{p}(\mathbf{x})$, denoted $\mathbf{s}(\tilde{\mathbf{x}})$, using the following equation, where the $d$-th index of $\mathbf{s}(\tilde{\mathbf{x}})$ is defined as

$$\mathbf{s}(\tilde{\mathbf{x}})_d = \frac{p_{\text{data}}(\bar{\mathbf{x}}_d + 1) - p_{\text{data}}(\bar{\mathbf{x}}_d)}{p_{\text{data}}(\bar{\mathbf{x}}_d + 1) * (\tilde{\mathbf{x}}_d - \bar{\mathbf{x}}_d) + p_{\text{data}}(\bar{\mathbf{x}}) * (\bar{\mathbf{x}}_d + 1 - \mathbf{x}_d)} \tag{19}$$

$$= \frac{\frac{p_{\text{data}}(\bar{\mathbf{x}}_d+1)}{p_{\text{data}}(\bar{\mathbf{x}}_d)} - 1}{\frac{p_{\text{data}}(\bar{\mathbf{x}}_d+1)}{p_{\text{data}}(\bar{\mathbf{x}}_d)} * (\tilde{\mathbf{x}}_d - \bar{\mathbf{x}}_d) + (\bar{\mathbf{x}}_d + 1 - \mathbf{x}_d)}, \tag{20}$$

where $\mathbf{x}_d \in \mathbb{Z}$ and $\mathbf{x}_d \in [\mathbf{x}_d, \mathbf{x}_d + 1)$. Thus, the Stein score $\mathbf{s}(\tilde{\mathbf{x}})$ of $\tilde{p}(\mathbf{x})$ can be constructed using the density ratio from Concrete scores. Given the recovered Stein score $\mathbf{s}(\hat{\mathbf{x}})$, we can perform Langevin dynamics to sample from $\tilde{p}(\mathbf{x})$, which gives the smoothed sample in Figure 2 (smoothed orange). We can then sample from $q(\mathbf{x}|\tilde{\mathbf{x}})$ using Equation (18) to perform closed-form denoising, which is shown in Figure 2 (orange).

### A.4 Connection to Stein Scores

In this section, we study a special case of the Concrete score, as described in Proposition 1, to highlight its connection to both the continuous (Stein) score and existing score matching methods. Given a $D$-dimensional continuous data distribution $p(\mathbf{x})$ and $\delta > 0$, we define a particular neighborhood structure $\mathcal{N}(\mathbf{x}) = \{\mathbf{x}_{n_i}\}_{i=1}^{D}$ where $\mathbf{x}_{n_i} = p(\mathbf{x} + \delta \mathbf{e}_i)$ and $\mathbf{e}_i$ is the standard (one-hot) basis vector with the $i$-th element $\mathbf{e}_i = 1$. Then:

$$\frac{\boldsymbol{c}_\theta(\mathbf{x}, \mathcal{N})}{\delta} = \frac{1}{p(\mathbf{x})} \left[ \frac{p(\mathbf{x} + \delta \mathbf{e}_1) - p(\mathbf{x})}{\delta}, \ldots, \frac{p(\mathbf{x} + \delta \mathbf{e}_D) - p(\mathbf{x})}{\delta} \right]^T \tag{21}$$

From Eq. 21, we can make two observations. First, we see that $\left[ \frac{p(\mathbf{x}+\delta \mathbf{e}_1)-p(\mathbf{x})}{\delta}, \ldots, \frac{p(\mathbf{x}+\delta \mathbf{e}_D)-p(\mathbf{x})}{\delta} \right]^T$ approximates the directional derivative of $p(\mathbf{x})$ via a first-order forward difference operation. As a result, the scaled Concrete score function $\frac{\boldsymbol{c}_\theta(\mathbf{x}, \mathcal{N})}{\delta}$ converges to $\boldsymbol{s}(\mathbf{x})$ in the limit of $\delta \to 0$:

$$\lim_{\delta \to 0} \frac{\boldsymbol{c}_\theta(\mathbf{x}, \mathcal{N})}{\delta} = \frac{1}{p(\mathbf{x})} \lim_{\delta \to 0} \left[ \frac{p(\mathbf{x} + \delta \mathbf{e}_1) - p(\mathbf{x})}{\delta}, \ldots, \frac{p(\mathbf{x} + \delta \mathbf{e}_D) - p(\mathbf{x})}{\delta} \right]^T = \frac{\nabla_{\mathbf{x}} p(\mathbf{x})}{p(\mathbf{x})} = \nabla_{\mathbf{x}} \log p(\mathbf{x})$$

which is precisely the definition of the Stein score.

## B  Additional Experimental Results

### B.1  Likelihood Evaluation on Discrete Tabular Data

We present additional results from Section 5.4, where we trained a MADE model using standard maximum likelihood. We use the training setting as detailed in Appendix C. As shown in Table 2, this maximum likelihood baseline serves as an upper bound on performance across all score-based methods. We note that as in continuous (Stein) score matching, log-likelihoods and score matching losses are not always correlated even though they both theoretically converge to the optimal solution given infinite model capacity [23]. This is due to practical constraints such as model mis-specification or optimization challenges. We believe that this is also the case for discrete score matching, which explains why directly optimizing likelihood outperforms all other approaches in Table 2.

### B.2  Neighborhood Structure Specification in Practice

In theory, our approach can use any neighborhood structure as long as the neighborhood-induced graph is connected (see Theorem 1). In practice, the choice of neighborhood structure can affect

| Datasets | Ratio Matching (↑) | Discrete Marginalization (↑) | CSM (Ours) (↑) | Log-Likelihood (↑) |
|---|---|---|---|---|
| NLTCS | -6.15 | -6.21 | **-6.13** | -6.00 |
| Plants | -15.44 | -19.03 | **-14.02** | -12.52 |
| Jester | -56.49 | -57.06 | **-54.91** | -51.77 |
| Amazon Diaper | **-10.69** | -42.52 | -11.13 | -9.82 |
| Amazon Feeding | **-12.09** | -35.96 | -12.65 | -11.29 |
| Amazon Gifts | -4.57 | -4.28 | **-4.22** | -3.43 |
| Amazon Media | **-10.22** | -13.77 | -10.30 | -7.79 |
| Amazon Toys | -9.83 | -16.34 | **-9.30** | -7.71 |

Table 2: Log-likelihood comparisons on discrete discrete tabular datasets. Higher is better. We find that CSM demonstrates good performance, almost always outperforming or performing comparably relative to the `Ratio Matching` and `Discrete Marginalization` baselines.

performance due to challenges in optimization and proper model specification. We provide an empirical analysis to build intuition on a series of 1-D synthetic datasets, and believe that the same intuition can generalize to higher dimensions.

First, we consider the 1-D distribution in Figure 5, where the data distribution does not contain any low density regions (regions with close to zero density). We parameterize the unnormalized probability distribution with softmax logits, and train the model using CSM. For the training objective, we explore 4 different neighborhood structures (3 different cycles as well as a fully-connected graph) as shown in Figure 5. In this setting, we observe that different neighborhood structures yield similar performances as evaluated by log-likelihood (see Figure 5).

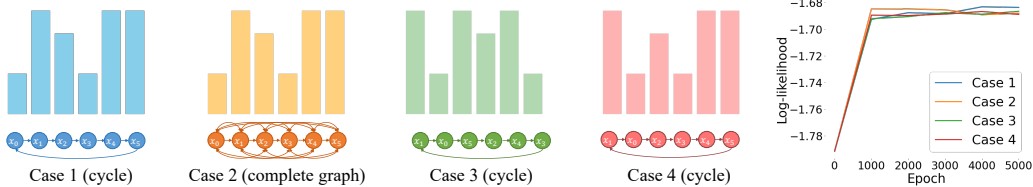

Figure 5: Examples of neighborhood structures and their corresponding log-likelihood values (higher is better) when trained with Concrete-SM.

Next, we consider a different 1-D data distribution which contains low density regions. As shown in Figure 6, we observe that the neighborhood structure in this setting plays a critical role in the final log-likelihoods: the complete graph in Case 2 outperforms all other structures. Interestingly, when the low-density regions are in between sets of high-density modes (as in Case 3 in Figure 6), we find that the model can still perform comparably relative to Case 2.

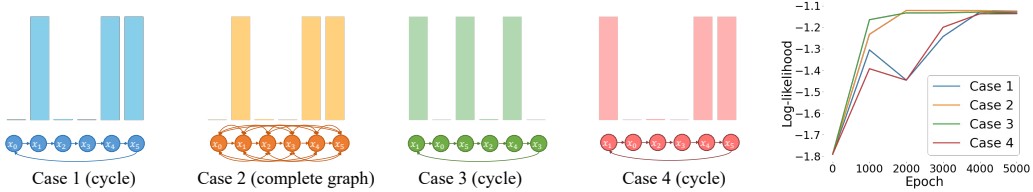

Figure 6: Examples of neighborhood structures and their corresponding log-likelihood values (higher is better) when trained with CSM.

This observation is similar to that of Stein score matching [26, 21]. Therefore, we believe that a similar noise annealing procedure followed by denoising CSM may potentially alleviate the effects of poorly chosen neighborhood structures for data distributions with low density regions. We will leave this investigation for future work. Such empirical results highlight that the best-performing structure in practice is often data-dependent. That is, a neighborhood graph which respects the particular dataset structure will perform the best out of all possible graph structures (analogous to the way in particular types of inductive biases are useful for solving specific tasks).

One interesting avenue for future research would be to draw inspiration from [49] and investigate whether it is possible to construct an "optimal" neighborhood structure for a given dataset. In this work, we uniformly sample a set of neighbors from $\mathcal{N}(\mathbf{x})$ when computing the CSM objective in practice (Algorithms 1 and 2), which can lead to issues with high variance during training. We hypothesize that an optimal proposal distribution over $\mathcal{N}(\mathbf{x})$ that minimizes variance may lead to insights on what the optimal neighborhood structure should be.

### B.3  Denoising Concrete Score Matching

We present additional experiments on denoising concrete score matching (D-CSM) as introduced in Section 4.3. In particular, we consider two 2-D toy benchmark datasets as commonly used in the density estimation literature [40, 41]. We quantize the data into $91 \times 91$ equally-distanced bins to obtain the discrete training data (see `Clean data` in Figure 7). Similar to our previous experiments, we use the grid neighborhood structure as detailed in Section 5.3 for training via D-CSM.

For the discrete noise distribution $\tilde{q}(\tilde{\mathbf{x}}|\mathbf{x})$, we consider the following distribution where each marginal is an independent 1-D Categorical distribution defined as:

$$\tilde{q}(\tilde{\mathbf{x}}|\mathbf{x})_i = \begin{cases} w & \text{when } \tilde{\mathbf{x}}_i = \mathbf{x}_i \\ \frac{1-w}{90} & \text{when } \tilde{\mathbf{x}}_i \neq \mathbf{x}_i, \end{cases} \tag{22}$$

where $0 < w < 1$ and $\tilde{q}(\tilde{\mathbf{x}}|\mathbf{x})_i$ denotes the $i$-th entry of $\tilde{q}(\tilde{\mathbf{x}}|\mathbf{x})$. Note that higher values of $w$ correspond to smaller noise levels. We use the same experimental setting as detailed in Appendix C.2, where we parameterize the un-normalized probability distribution with softmax logits, but train the model using D-CSM (see Equation (6)).

We present the results in Figure 7. We observe that samples from our model (green) matches samples from the ground truth noisy data distributions $\tilde{p}(\tilde{\mathbf{x}})$ (blue), indicating the practical effectiveness of D-CSM. We leave a more in-depth exploration of the D-CSM approach for future work.

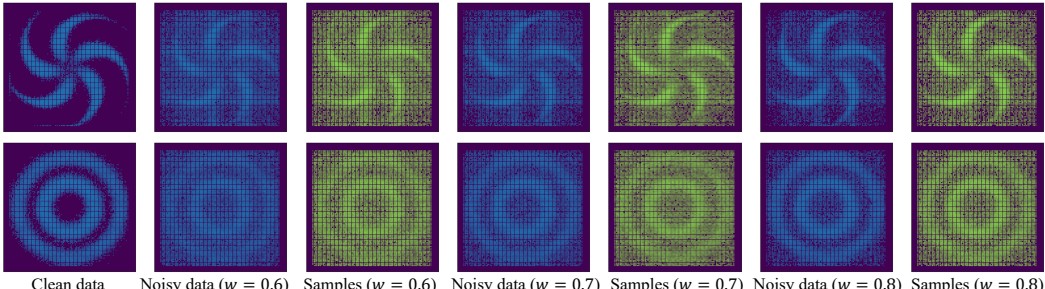

Clean data  Noisy data ($w = 0.6$)  Samples ($w = 0.6$)  Noisy data ($w = 0.7$)  Samples ($w = 0.7$)  Noisy data ($w = 0.8$)  Samples ($w = 0.8$)

Figure 7: Samples from models trained with D-CSM. The blue samples are the ground truth samples from $\tilde{p}(\tilde{\mathbf{x}})$ obtained by perturbing the clean data distribution with $\tilde{q}(\tilde{\mathbf{x}}|\mathbf{x})$. As we increase $w$, the variance of the noise distribution $\tilde{q}(\tilde{\mathbf{x}}|\mathbf{x})$ decreases and the perturbed data distribution $\tilde{p}(\tilde{\mathbf{x}})$ becomes less noisy. The samples from our models (green) match the ground truth perturbed data samples (blue) across different values of $w$, demonstrating the effectiveness of D-CSM.

## C  Experimental Details

In this section, we provide more details for each of the experimental settings. We will release our code upon publication.

### C.1  1-D Discrete Data

We discuss the experimental setting for Section 5.2. We use a 16-category 1-D data distribution with class labels from 0 to 15, and normalize the data to $[0, 1]$ before feeding it into the model. We use a 3-layer shallow MLP for the score network with 100 hidden units and Tanh activations. We use

the model to directly parameterize the Concrete score and define the neighborhood structure to be a Cycle structure as shown in Figure 8. We train the model using the Adam optimizer with learning rate 0.001 for 100,000 iterations. The Langevin samples and denoised samples in Figure 2 are obtained using the approach detailed in Appendix A.3.

## C.2  2-D Multi-Class Datasets

For the 2-D experiments, we consider three 2-D toy benchmark datasets with multiple modes and discontinuities as commonly used in the density estimation literature [40, 41]. We quantize the data into $91 \times 91$ equally-distanced bins to obtain the discrete training data (see Figure 3). We use the grid neighborhood structure for training via CSM (see Figure 8). For a fair comparison, we use the same model architecture and training configurations across all methods. We directly parameterize the unnormalized probability distribution with softmax logits, but train the model using the baselines and CSM. Since there are $91 \times 91$ possible classes, our model directly parameterizes an unnormalized distribution with $91 \times 91$ possible values. We initialize the model to have a uniform probability across classes and train the model using the Adam optimizer with learning rate 0.0005. To sample from the model, we can either use likelihood sampling (by normalizing the model) or Metropolis-Hastings. We empirically observe that both approaches work well.

## C.3  Discrete Tabular Data

For the tabular experiments, we consider tabular (discrete) datasets drawn from both the Twenty Datasets [43] and the Amazon Baby Registries benchmarks [44]. In order to evaluate likelihoods, we directly parameterize the probability distribution with a discrete autoregressive model (MADE [45]), but train the model using the baseline approaches and CSM. The MADE model uses 2 residual blocks with latent dimension 100 and Tanh activations. We train the model for 50000 iterations using the Adam optimizer with learning rate 0.0005. We use a grid neighborhood structure for CSM (see Figure 8). For a fair comparison, we use the same model architecture and experimental configurations across all methods. Similar to our previous experiments, we use the grid neighborhood structure for training via CSM (see Figure 8).

## C.4  Discrete Image Data

We experiment with the binarized MNIST [46] dataset, which has 784 dimensions. We perturb the images $\mathbf{x}$ with binarized Gaussian noise $q(\tilde{\mathbf{x}}|\mathbf{x}) = \mathcal{N}(\tilde{\mathbf{x}}|\mathbf{x}, \sigma_i^2 I)$ ($i = 1, ..., 7$). Specifically, given a sample from the noisy data distribution $\tilde{\mathbf{x}} \sim q(\tilde{\mathbf{x}}|\mathbf{x})$, we discretize $\tilde{\mathbf{x}}$ by mapping it to its nearest integer neighbor, and then take the modula by 2 to map $\tilde{\mathbf{x}}$ into binary bins. We note that this process is similar to applying Categorical noise. In our experiment, we use $\{\sigma_1, \sigma_2, \sigma_3, \sigma_4, \sigma_5, \sigma_6, \sigma_7\} = \{0.1, 0.2, 0.25, 0.3, 0.4, 0.5, 0.65\}$. We normalize the data to lie in between $[-1, 1]$ before feeding it into the model. We directly parameterize the Concrete score function with a U-Net [47], and train the model using CSM based on a grid neighborhood structure. We train the models until convergence with the Adam optimizer using learning rate 0.0002.

We use Metropolis-Hastings for sampling from the model after training: in particular, we initialize the sample to be drawn from uniform binary noise then use the model corresponding to noise level $\sigma_7$ to perform Metropolis-Hastings updates. After that, we use the output from the model to initialize the input for sampling from the model with noise level $\sigma_6$. We then repeat the procedure and initialize the model corresponding to the previous noise level with the output from the next noise level—a process similar to [26]. We observe that each noise level requires around 100-800 steps to obtain reasonable results.

## C.5  Training for Special Neighborhood Structures

In this section, we provide additional details on special neighborhood structures where we can design even more efficient training procedures than using Algorithm 1 and Algorithm 2. For simplicity, we name the elements in $\mathcal{X}$: $\{\mathbf{x}_1, \mathbf{x}_2, ..., \mathbf{x}_N\}$.

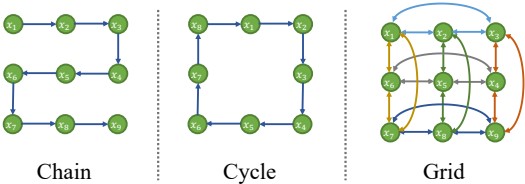

Figure 8: Special types of neighborhood structures considered in the paper.

**Chain.** When the neighborhood-induced graph is a Chain (Figure 8), the second term $\mathcal{J}_2$ in Equation (5) simplifies to the following via reparameterization:

$$\sum_{k=1}^{N-1} 2p_{\text{data}}(\mathbf{x}_{k+1})\boldsymbol{c}_\theta(\mathbf{x}_k;\mathcal{N})_1 = \sum_{k=2}^{N} 2p_{\text{data}}(\mathbf{x}_k)\boldsymbol{c}_\theta(\mathbf{x}_{k-1};\mathcal{N})_1. \tag{23}$$

Note that the subscript of $\boldsymbol{c}_\theta$ is always one since each node has at most one neighbor.

**Cycle.** Similarly, when the neighborhood-induced graph is a Cycle (Figure 8), the second term $\mathcal{J}_2$ becomes:

$$\sum_{k=1}^{N} 2p_{\text{data}}(\mathbf{x}_{k+1})\boldsymbol{c}_\theta(\mathbf{x}_k;\mathcal{N})_1 = \sum_{k=1}^{N} 2p_{\text{data}}(\mathbf{x}_k)\boldsymbol{c}_\theta(\mathbf{x}_{(k-1\,\text{mod}N)};\mathcal{N})_1 \tag{24}$$

Thus for both Chains and Cycles, the term $\mathcal{J}_2$ in Equation (5) can be evaluated efficiently using samples from $p_{\text{data}}(\mathbf{x})$.

**Grid.** When the neighborhood-induced graph is a Grid (Figure 8), the second term $\mathcal{J}_2$ can be approximated efficiently by decomposing each dimension into smaller Cycles. Specifically, to estimate $\mathcal{J}_2$, we can uniformly sample a dimension $d$, and then use the same reparameterization approach used for the Cycle structure at each dimension.

# D  Additional Related Works and Corrections

In this section, we provide additional details about the baseline methods: in particular, we found that the original equations in [34] were incorrect. We elaborate upon them below.

## D.1  Ratio Matching with Discrete Data

We assume with slight abuse of notation that $\mathcal{X} \in \{0, 1\}^D$, and $p_{\text{data}}(\mathbf{x})$ is the unknown discrete data distribution. Ratio matching [42, 34] was originally proposed as an alternative approach to score matching for learning discrete binary probability distributions from samples. Similar to score matching, it leverages the fact that ratios of probabilities are also independent of the intractable normalizing constant (due to cancellation), and seeks to match the ground truth density ratios $\frac{p_{\text{data}}(\mathbf{x})}{p_{\text{data}}(\mathbf{x}_{-d})} = \frac{q_\theta(\mathbf{x})}{q_\theta(\mathbf{x}_{-d})}$ where $\mathbf{x}_{-d}$ denotes the vector $\mathbf{x}$ with the $d$th entry bit-flipped (e.g. from 0 to 1). Concretely, ratio matching minimizes the following objective:

$$\mathcal{J}_{RM}(\theta) = \tfrac{1}{2}\mathbb{E}_{p_{\text{data}}(\mathbf{x})}\left[\sum_{d=1}^{D}\left(g\left(\frac{p_{\text{data}}(\mathbf{x})}{p_{\text{data}}(\mathbf{x}_{-d})}\right) - g\left(\frac{q_\theta(\mathbf{x})}{q_\theta(\mathbf{x}_{-d})}\right)\right)^2 + \left(g\left(\frac{p_{\text{data}}(\mathbf{x}_{-d})}{p_{\text{data}}(\mathbf{x})}\right) - g\left(\frac{q_\theta(\mathbf{x}_{-d})}{q_\theta(\mathbf{x})}\right)\right)^2\right] \tag{25}$$

where $g(z) = \frac{1}{1+z}$ for $z \in \mathbb{R}^+$ is a nonlinear transformation of the ratios for numerical stability. The symmetrized objective tries to minimize the squared distance between the two density ratios.

As the original ratio matching approach [42, 34] is proposed for binary data, [34] extends ratio matching to multi-class data. Following the notation in [34], let $\mathbf{x}^{\backslash i}$ denote the vector formed by dropping the $i$-th element (which we call $\mathbf{x}_i$) from $\mathbf{x}$. To make things concrete, this means that $p(\mathbf{x}) = p(\mathbf{x}^{\backslash i}, \mathbf{x}_i)$. We also let $q_\theta(\xi_i|\mathbf{x}^{\backslash i})$ denote the conditional probability for the $i$-th index of $\mathbf{x}$ taking the value $\xi_i$, and let $q_\theta(\sim \xi_i|\mathbf{x}^{\backslash i})$ denote the conditional probability for the $i$-th index of $\mathbf{x}$ not taking the value $\xi_i$.

The generalized ratio matching objective proposed in [34] is:

$$\theta^* = \arg\min_\theta \sum_{\mathbf{x}} p_{\text{data}}(\mathbf{x}) \sum_{i=1}^{d}\sum_{\xi_i}\left[g\left(\frac{p(\xi_i, \mathbf{x}^{\backslash i})}{p(\sim \xi_i, \mathbf{x}^{\backslash i})}\right) - g\left(\frac{q_\theta(\xi_i, \mathbf{x}^{\backslash i})}{q_\theta(\sim \xi_i, \mathbf{x}^{\backslash i})}\right)\right]^2 \tag{26}$$

which can be simplified as:

$$\theta^* = \arg\min_\theta \sum_{\mathbf{x}} p_{\text{data}}(\mathbf{x}) \sum_{i=1}^{d}\sum_{\xi_i}(1 - q_\theta(\xi_i|\mathbf{x}^{\backslash i}))^2. \tag{27}$$

It is easy to see that the optimal $\theta^*$ for Equation (27) can be achieved independent of $p_{\text{data}}(\mathbf{x})$, which is problematic. We also empirically show that model trained with Equation (27) cannot learn meaningful features based on the data (see `Ratio` in Figure 3).

As an alternative, we provide a corrected multi-class ratio matching objective. Similar to [42, 48], we assume that all the probabilities are non-zero. For simplicity, we use $p(\mathbf{x})$ to denote $p_{\text{data}}(\mathbf{x})$. The objective for multi-class ratio matching can achieved by optimizing:

$$\theta^* = \arg\min_\theta \sum_{\mathbf{x}} p(\mathbf{x}) \sum_{i=1}^{d}\sum_{\xi_i}\left[g\left(\frac{p(\xi_i, \mathbf{x}^{\backslash i})}{p(\sim \xi_i, \mathbf{x}^{\backslash i})}\right) - g\left(\frac{q_\theta(\xi_i, \mathbf{x}^{\backslash i})}{q_\theta(\sim \xi_i, \mathbf{x}^{\backslash i})}\right)\right]^2 \tag{28}$$

$$= \arg\min_\theta \sum_{\mathbf{x}} p(\mathbf{x}) \sum_{i=1}^{d}\left[(1 - q_\theta(\mathbf{x}_i|\mathbf{x}^{\backslash i}))^2 + \sum_{\xi_i \neq \mathbf{x}_i} q_\theta(\xi_i|\mathbf{x}^{\backslash i})^2\right]. \tag{29}$$

*Proof.*

$$\theta^* = \arg\min_\theta \sum_{\mathbf{x}} p(\mathbf{x}) \sum_{i=1}^{d} \sum_{\xi_i} \left[ g\left( \frac{p(\xi_i, \mathbf{x}^{\backslash i})}{p(\sim \xi_i, \mathbf{x}^{\backslash i})} \right) - g\left( \frac{q_\theta(\xi_i, \mathbf{x}^{\backslash i})}{q_\theta(\sim \xi_i, \mathbf{x}^{\backslash i})} \right) \right]^2$$

$$= \arg\min_\theta \sum_{\mathbf{x}} p(\mathbf{x}) \sum_{i=1}^{d} \sum_{\xi_i} \left[ p(\sim \xi_i | \mathbf{x}^{\backslash i}) - q_\theta(\sim \xi_i | \mathbf{x}^{\backslash i}) \right]^2$$

$$= \arg\min_\theta \sum_{\mathbf{x}} p(\mathbf{x}) \sum_{i=1}^{d} \sum_{\xi_i} \left[ p(\xi_i | \mathbf{x}^{\backslash i}) - q_\theta(\xi_i | \mathbf{x}^{\backslash i}) \right]^2$$

$$= \arg\min_\theta \sum_{\mathbf{x}} p(\mathbf{x}) \sum_{i=1}^{d} \sum_{\xi_i} \left[ q_\theta^2(\xi_i | \mathbf{x}^{\backslash i}) - 2 p(\xi_i | \mathbf{x}^{\backslash i}) q_\theta(\xi_i | \mathbf{x}^{\backslash i}) \right]^2$$

$$= \arg\min_\theta \sum_{\mathbf{x}} p(\mathbf{x}) \sum_{i=1}^{d} \sum_{\xi_i} q_\theta^2(\xi_i | \mathbf{x}^{\backslash i}) - \sum_{\mathbf{x}} p(\mathbf{x}) \sum_{i=1}^{d} \sum_{\xi_i} 2 p(\xi_i | \mathbf{x}^{\backslash i}) q_\theta(\xi_i | \mathbf{x}^{\backslash i})$$

$$= \arg\min_\theta \sum_{\mathbf{x}} p(\mathbf{x}) \sum_{i=1}^{d} \sum_{\xi_i} q_\theta^2(\xi_i | \mathbf{x}^{\backslash i}) - \sum_{i=1}^{d} \sum_{\mathbf{x}^{\backslash i}, \mathbf{x}_i} p(\mathbf{x}) \sum_{\xi_i} 2 p(\xi_i | \mathbf{x}^{\backslash i}) q_\theta(\xi_i | \mathbf{x}^{\backslash i})$$

$$= \arg\min_\theta \sum_{\mathbf{x}} p(\mathbf{x}) \sum_{i=1}^{d} \sum_{\xi_i} q_\theta^2(\xi_i | \mathbf{x}^{\backslash i}) - \sum_{i=1}^{d} \sum_{\mathbf{x}^{\backslash i}, \xi_i} 2 p(\xi_i, \mathbf{x}^{\backslash i}) q_\theta(\xi_i | \mathbf{x}^{\backslash i})$$

$$= \arg\min_\theta \sum_{\mathbf{x}} p(\mathbf{x}) \sum_{i=1}^{d} \sum_{\xi_i} q_\theta^2(\xi_i | \mathbf{x}^{\backslash i}) - \sum_{i=1}^{d} \sum_{\mathbf{x}} 2 p(\mathbf{x}) q_\theta(\mathbf{x}_i | \mathbf{x}^{\backslash i})$$

$$= \arg\min_\theta \sum_{\mathbf{x}} p(\mathbf{x}) \sum_{i=1}^{d} \frac{\sum_{\xi_i} q_\theta^2(\xi_i, \mathbf{x}^{\backslash i}) + 2 q_\theta(\mathbf{x}_i, \mathbf{x}^{\backslash i}) q_\theta(\mathbf{x}^{\backslash i})}{q_\theta^2(\mathbf{x}^{\backslash i})}$$

$$= \arg\min_\theta \sum_{\mathbf{x}} p(\mathbf{x}) \sum_{i=1}^{d} \frac{-q_\theta^2(\mathbf{x}^{\backslash i}) + \sum_{\xi_i \neq \mathbf{x}_i} q_\theta^2(\xi_i, \mathbf{x}^{\backslash i}) + (\sum_{\xi_i \neq \mathbf{x}_i} q_\theta(\xi_i, \mathbf{x}^{\backslash i}))^2}{q_\theta^2(\mathbf{x}^{\backslash i})}$$

$$= \arg\min_\theta \sum_{\mathbf{x}} p(\mathbf{x}) \sum_{i=1}^{d} \frac{\sum_{\xi_i \neq \mathbf{x}_i} q_\theta^2(\xi_i, \mathbf{x}^{\backslash i}) + (q_\theta(\mathbf{x}^{\backslash i}) - q_\theta(\mathbf{x}_i, \mathbf{x}^{\backslash i}))^2}{q_\theta^2(\mathbf{x}^{\backslash i})}$$

$$= \arg\min_\theta \sum_{\mathbf{x}} p(\mathbf{x}) \sum_{i=1}^{d} \left[ (1 - q_\theta(\mathbf{x}_i | \mathbf{x}^{\backslash i}))^2 + \sum_{\xi_i \neq \mathbf{x}_i} q_\theta^2(\xi_i | \mathbf{x}^{\backslash i}) \right].$$

$\square$

We provide the samples trained with Equation (28) in Figure 3 (`Ratio-fixed`). We observe that the samples appear to be more reasonable then those from a model trained with Equation (27) (`Ratio`).

### D.2 Discrete Marginalization

Given a multi-class discrete data distribution $p_{\text{data}}(\mathbf{x})$, discrete marginalization [34] proposes to learn the data distribution by minimizing

$$\sum_{\mathbf{x}} p_{\text{data}}(\mathbf{x}) \sum_{i=1}^{d} \left[ \left( \frac{\mathcal{M}_i p}{p} \right)^2 + \left( \frac{\mathcal{M}_i q_\theta}{q_\theta} \right)^2 - 2 \mathcal{M}_i \left( \frac{\mathcal{M}_i q_\theta}{q_\theta} \right) \right], \tag{30}$$

where $\mathcal{M}_i$ is the marginalization operator defined as $\mathcal{M}_i : \mathcal{F}^1 \to \mathcal{F}^1 : f(\mathbf{x}) \to \int_{\mathbf{x}_i} f(\mathbf{x}) d\mathbf{x}_i$ or $\sum_{\mathbf{x}_i} f(\mathbf{x})$ in the discrete case. Thus $\frac{\mathcal{M}_i p}{p} = \frac{1}{p_{\text{data}}(\mathbf{x}_i | \mathbf{x}^{\backslash i})}$ and $\frac{\mathcal{M}_i q_\theta}{q_\theta} = \frac{1}{q_\theta(\mathbf{x}_i | \mathbf{x}^{\backslash i})}$.

The authors further simplified Equation (30) to

$$\theta^* = \arg\min_\theta \sum_{\mathbf{x}} p_{\text{data}}(\mathbf{x}) \sum_{i=1}^d \sum_{\xi_i} \frac{1 - 2q_\theta(\xi_i|\mathbf{x}^{\setminus i})}{q_\theta^2(\xi_i|\mathbf{x}^{\setminus i})}. \tag{31}$$

However, we observe that this is a typo in Equation (31)—the optimal $\theta^*$ can be achieved while being independent of $p_{\text{data}}(\mathbf{x})$. In our experiments, we also observe that model trained with Equation (31) cannot learn meaningful representations of the data (see `Marginal` in Figure 3).

In the following, we provide the corrected simplified version of Equation (30).

**Theorem 5.** *Optimizing Equation* (30) *is equivalent to optimizing*

$$\theta^* = \arg\min_\theta \sum_{\mathbf{x}} p_{\text{data}}(\mathbf{x}) \sum_{i=1}^d \left[ \frac{1}{q_\theta^2(\mathbf{x}_i|\mathbf{x}^{\setminus i})} - \sum_{\xi_i} \frac{2}{q_\theta(\xi_i|\mathbf{x}^{\setminus i})} \right]. \tag{32}$$

*Proof.*

$$\theta^* = \arg\min_\theta \sum_{\mathbf{x}} p_{\text{data}}(\mathbf{x}) \sum_{i=1}^d \left[ \left( \frac{\mathcal{M}_i p}{p} \right)^2 + \left( \frac{\mathcal{M}_i q_\theta}{q_\theta} \right)^2 - 2\mathcal{M}_i \left( \frac{\mathcal{M}_i q_\theta}{q_\theta} \right) \right] \tag{33}$$

$$= \arg\min_\theta \sum_{\mathbf{x}} p_{\text{data}}(\mathbf{x}) \sum_{i=1}^d \left[ \left( \frac{\mathcal{M}_i q_\theta}{q_\theta} \right)^2 - 2\mathcal{M}_i \left( \frac{\mathcal{M}_i q_\theta}{q_\theta} \right) \right] \tag{34}$$

$$= \arg\min_\theta \sum_{\mathbf{x}} p_{\text{data}}(\mathbf{x}) \sum_{i=1}^d \left[ \frac{1}{q_\theta^2(\mathbf{x}_i|\mathbf{x}^{\setminus i})} - \sum_{\xi_i} \frac{2}{q_\theta(\xi_i|\mathbf{x}^{\setminus i})} \right]. \tag{35}$$

In the last equation, we used the fact that $\sum_{\mathbf{x}} p_{\text{data}}(\mathbf{x}) \mathcal{M}_i(\frac{\mathcal{M}_i q_\theta}{q_\theta}) = \sum_{\mathbf{x}} p_{\text{data}}(\mathbf{x}) \sum_{\xi_i} (\frac{\mathcal{M}_i q_\theta}{q_\theta}) = \sum_{\mathbf{x}} p_{\text{data}}(\mathbf{x}) \sum_{\xi_i} \frac{1}{q_\theta(\xi_i|\mathbf{x}^{\setminus i})}$, which directly follows from Lemma 4 in [34].

□

We provide the samples trained with Equation (32) in Figure 3 (`Marginal-fixed`). We observe that the samples appear to be more reasonable then those from a model trained with Equation (31) (`Marginal`).