# OpenReview forum: "Concrete Score Matching: Generalized Score Matching for Discrete Data"
_NeurIPS.cc/2022/Conference — NeurIPS 2022 Accept_

### Official Review · Reviewer_PXyS · 2022-07-10

**Rating:** 6
**Confidence:** 3
**Soundness:** 3 good
**Presentation:** 3 good
**Contribution:** 3 good

**Summary:**

This paper proposes a comprehensive framework of score matching on discrete data. The papers proposes a clever surrogate of the score function, and proves its completeness for practical usage. After establishing the concrete score, this paper proposes the concrete score matching to estimate it, as well as a denoising version. To leverage the concrete score, this paper also proposes a MCMC algorithm to sample based on the score. Experiments show effectiveness of the concrete score matching.

**Questions:**



**Limitations:**



**Strengths And Weaknesses:**

Strengths:

The author carefully extends the score matching to discrete data. This extension finds counterparts of continuous score matching in both theory (i.e., the completeness), training (i.e., concrete score match and its denoising version), inference (i.e., MCMC). It also allows directly estimate the concrete score, instead of the probability mass function. It seems many benefits in the continuous score matching also exist in the concrete score matching.

Weakness \& Questions:

1. Why $J_1$ requires Monte Carlo on $i$? It seems that the summation on $i$ can be directly calculated, since $c_\theta(x; N)$ is the output of the neural network, and simply summing each dimension of the (square of) output is exactly the summation on $i$.

2. In Line 221, the author claims that the concrete score $c_{\tilde{p}}(\tilde{x}; N)$ captures how close $\tilde{x}$ is to x. However, the concrete score is only related to $\tilde{x}$? How to understand the closeness?

3. Does the concrete score of $\tilde{q}(\tilde{x}|x)$ have a closed form expression as the continuous case?

4. Is the neighbor in L 203 and L 220 suitable choices? Since $|N(x)|$ = $2^D$ and $3^D$ respectively, which are too large to model using a neural network when $D$ is large.

5. The author also uses annealed noise as [21]. Is the concrete score matching able to scale up to more complex data as [21]? For example, CIFAR10 and ImageNet. If not, what is the gap between this work and [21]?

6. Missing related works on score matching: [1*] and [2*] are related works, which also enlarge the scope of score matching.

[1*] Bao et al., Bi-level score matching for learning energy-based latent variable models

[2*] Bao et al., Variational (gradient) estimate of the score function in energy-based latent variable models

---

> ### Author Response · Authors · 2022-08-01
> **Clarified the writing and added discussion on additional references**
>
> **Q: Clarification of J1.**
>
> **A:** As referenced in l. 192 (as well as in the general comment), the summation requires the evaluation over all |N(x)| neighbors, which can be expensive. Therefore, we use Monte Carlo sampling over the neighbors to obtain an unbiased estimate of the gradient for computing the loss.
>
>
> **Q: “Closeness of x tilde”.**
>
> **A:** The denoising variant of the concrete score $c_{\tilde{p}}(\tilde{x})(\tilde{x}; \mathcal{N})$ relates both the corrupted data point $\tilde{x}$ and the clean input $x$ via the expression in l. 220. That is, the posterior distribution of the clean input given the perturbed input is used to obtain $c_{\tilde p}$. Then, the closeness of tilde x and x can be understood analogously to the continuous variant of DSM, where the magnitude of D-CSM will be larger for dissimilar pairs.
>
>
> **Q: Scalability of neighbors in L 203 and L 220.**
>
> **A:** As also mentioned in the general response, although we use Monte Carlo approximations to approximate the sum over all neighbors in Theorem 3 for high dimensional datasets, we provide efficient training objectives for the loss in Algorithms 1 and 2. This is exactly why it is feasible to use the neighborhood structure specified in l. 203 and 220 in the original text (l. 211 and 228 in the revised PDF) . These approximations are similar in spirit to sliced score matching (SSM) (Song et al. 2019) and we do not observe issues with high variance in practice. In particular for our MNIST experiments, we found that such slicing/projection steps (sampling by coordinate) worked well.
>
>
> **Q: Closed form of denoised CSM.**
>
> **A:** Given the noise distribution $\tilde{q}(\tilde{x}| x)$ and the neighborhood structure $\mathcal{N}$, the same argument for continuous (Stein) scores holds for Concrete scores. That is, given any neighbor $\tilde x_{n_i}\in \mathcal{N}(\tilde x)$, the i-th index of the concrete score $c_{\tilde{q}(\tilde x|x)}(x;\mathcal{N})$ can be computed in closed form $\frac{\tilde{q}(\tilde x_{n_i}|x)- \tilde{q}(\tilde x|x)}{\tilde{q}(\tilde x|x)}$ as defined in Eq (2). The exact form of this expression will depend on the probability mass function of the noise distribution.
>
>
> **Q: “What is the gap between this work and [21]?”**
>
> **A:** Although we also present a denoising variant of CSM with annealed noise as in [21], our main focus is to propose a new “score matching” objective (CSM) for discrete data. Unlike our work, [21] focuses on training a sequence of hierarchical models using the existing denoisng score matching (DSM). We believe the exploration of training hierarchical models using CSM is not the main focus of our current work but would be a meaningful future direction.
>
> We believe the gap between CSM and [21] is largely empirical—in order to scale up our method to even more complex datasets such as ImageNet, we would require more practical experimentation such as: (1) exploring model architectures that may be better suited for discrete score matching; (2) better ways of handling low-density regions in discrete distributions (specifying neighborhood structures) as in the additional results we provided in Appendix B.2; (3) exploring noise distributions and annealing schedules; among others. We believe all of these directions play a crucial role in the success of [21] and will require a significant amount of modifications when applied to discrete data. As the main goal of this work is to propose a new “score matching” approach for discrete data, we leave these explorations to future work.
>
> **Q: Additional references.**
>
> **A:** Thank you for the 2 additional references – we have included them in Section 6.

---

> > ### Comment · Reviewer_PXyS · 2022-08-07
> > **Thanks**
> >
> > Thanks for the response, which addresses some of my confusion. I'd keep my score.

---

### Official Review · Reviewer_rPQu · 2022-07-10

**Rating:** 5
**Confidence:** 4
**Soundness:** 3 good
**Presentation:** 3 good
**Contribution:** 2 fair

**Summary:**

This paper proposes a discrete counterpart to continuous score
matching. The main idea is to build a neighboring graph for each point
and replace the gradient of the log-likelihood with a vector
containing the likelihood ratios between a datapoint and its
neighbors. This vector is called the concrete score. The paper
proposes a training scheme to learn the concrete score from data, and
shows that in the limit of infinite data and infinite capacity the
model can recover the true data likelihood. An efficient
implementation is demonstrated empirically on simple discrete datasets.


**Questions:**

Please see the weaknesses points above.

**Limitations:**

The authors do mention the limitations in scaling to higher dimensions and larger datasets. I agree this is the main limitation and the paper could be improved by conveying a better understanding of what these limits are in practice.

**Strengths And Weaknesses:**

Strengths:

1) The paper is well written and the ideas are clearly explained.
2) The proposed concrete score is simple and elegantly converges to the
  Stein score in the limit of neighboring graph shrinking.
3) The paper derives a training objective that is in the same spirit as
  in continuous score matching.

Weaknesses:

1) I found the paper dedicates very little attention to how the
   neighborhoods are actually built in practice for the different
   datasets. In particular for the high-dimensional datasets, this
   seems to be a crucial aspect of the method that was disregarded
   (unless I missed something).
2) The method has scaling issues for high-dimensional data and large
  datasets. Theorem 1 requires a fully connected graph which is
  impractical for large datasets, and the approximated loss will
  certainly suffer from high variance. Although the authors mentioned
  this, it is not entirely clear what are the practical limitations
  for scaling up.
3) The experimental evaluation is limited. The paper only compares to
   two baseline methods. The gains do not seem significant, and the performance
   comparison was limited to log-likelihood, without taking efficiency into account. These
   baseline methods were unknown to me, and I felt more detail was
   required to understand what is being compared to.  I wonder how the method would compare to other
   discrete generative models such as D3PM [Austin et al.], [43], to
   name a few, and in particular the very related method of finite
   differences FSDM [27].

---

> ### Author Response · Authors · 2022-08-01
> **CSM is scalable to high-dimensional data. Extra analysis on neighborhood structure is added.**
>
> **Q: Practical selection of neighborhood structure.**
>
> **A:** We added Appendix B.2 in the revised PDF (Supplementary Material) with additional experiments on synthetic datasets to elaborate upon this point. In particular, we experimented with 4 different neighborhood structures: a complete graph, as well as 3 different orderings of nodes in a cycle. In a setting where the data distribution is well-behaved (does not contain any low density regions), we found that the neighborhood structure did not play a significant role in the model’s final performance as evaluated by log-likelihood—all structures performed similarly (Figure 5 in Appendix B.2). However, when we modified the data distribution to be more ill-conditioned (i.e., contain low-density regions between high density modes), a particular cycle configuration as well as the complete graph converged faster than all other approaches (Figure 6 in Appendix B.2). This observation shares similarity with Stein score optimization ([21] Song et al.), where low-density regions in the data distribution pose additional challenges to optimization and learning in practice (and in our case neighborhood selection).
>
> Such results emphasize that while the neighborhood structure can be any connected structure in theory (Theorem 1), the best-performing structure in practice is often data-dependent. That is, a neighborhood graph which respects/takes into account the particular dataset structure will perform the best (analogous to the way in particular types of inductive biases are useful for solving specific tasks). We believe that in practice—especially for high-dimensional datasets—selecting the right neighborhood structure is a crucial aspect of CSM, and additional heuristics may be needed to design the ideal structure. We have also added a short discussion about this in Appendix B.2.
>
> **Q: Scalability to higher dimensions**
>
> **A:** Thank you for this question. As we mentioned in the general response, Theorem 1 does not require a fully-connected graph (i.e., a complete graph). As in Figure 1 (as well as in l. 98-106), we only require that the neighborhood structure is connected. This allows for the specification of a variety of graphs such as stars and chains. And although we use Monte Carlo approximations to approximate the sum over all neighbors in Theorem 3 for high dimensional datasets, we provide efficient training objectives for the loss in Algorithms 1 and 2. These approximations are similar in spirit to sliced score matching (SSM) (Song et al. 2019) for (Stein) score matching. In particular for our MNIST experiments, which is a much higher-dimensional dataset, we found that such slicing/projection steps (sampling by coordinate) worked well.
>
> We note that we did not run into practical issues with high variance in the training objective using the neighborhood structures we experimented with in the paper. However, in our new experiments in Appendix B.2 (Figures 5 & 6), we observed that certain neighborhood structures lead to faster convergence and better performance depending on the dataset. In general, we believe that this could be a bigger problem for more ill-conditioned data distributions and we leave this for future work.
>
>
> **Q: Comparison with additional baselines.**
>
> **A:** We thank the reviewer for mentioning this point, and we provide additional clarifications about the baselines that we compared against (Ratio Matching and Discrete Marginalization) in our experiments. As mentioned in the main text and also by R1, Ratio Matching and Discrete Marginalization are the most relevant to CSM. They are two ways of performing score matching for discrete distributions as suggested in a prior work (Lyu 2012). However, we’d like to emphasize that the approaches presented in (Lyu 2012) are not only computationally inefficient, but also less flexible than our CSM approach.
>
> In terms of the other generative modeling baselines of D3PM and FD-SM as suggested by R2, we highlight fundamental differences between them and our approach. (1) For D3PM, their contribution is orthogonal to our work: they present a diffusion probabilistic model for discrete data that must still be trained via (approximate) maximum likelihood training. However, our CSM approach presents a new score matching objective for discrete data that can scale to high-dimensional datasets. We have added this reference to Section 6. (2) For FD-SM, as mentioned in l. 142-148, CSM does not explicitly try to match the Stein score. Rather, we posit a particular neighborhood structure across the dataset dimensions to construct a surrogate gradient that allows us to perform score matching over discrete data.

---

> > ### Comment · Reviewer_rPQu · 2022-08-06
> > **Thank you!**
> >
> > Thank you very much for your detailed answers and for the improvements to
> > the manuscript. Apologies for my misunderstanding of Theorem 1.
> >
> > I think the paper has very solid and elegant theory, and the
> > presentation is of very high quality. Yet, I still remain a bit
> > skeptical about the scaling to higher dimensions.
> >
> > I appreciate the new material in appendix B discussing the
> > neighborhood structures. It seems reasonable that annealing would help
> > for unevenly distributed data, and hopefully make the method more
> > robust to the selected neighboring structure. Yet, except for the
> > complete or the grid graphs, it is still not entirely clear to me how
> > to define the cycle (or chain) graph in high-dimensional data, so
> > extrapolating these observations to high dimensions remains difficult.
> >
> > Moreover, even in the case of the grid graph, the neighborhood size
> > will obviously explode for real high-dimensional data. I appreciate
> > the binary MNIST experiments as a first approximation showing that the
> > proposed training procedure is feasible. I wish the paper also included a
> > quantitative comparison including the trade-offs in computational cost for this case.
> >
> > Overall, this is a well presented and theoretically elegant
> > method. Scalability to real high-dimensional data is not yet fully
> > demonstrated in my opinion, but I think the positive aspects currently
> > outweigh this point. I will therefore raise my score.

---

> > > ### Author Response · Authors · 2022-08-07
> > > **Additional points about the scalability of CSM to higher dimensions.**
> > >
> > > Thanks for taking the time to read through our response. We are glad that we were able to clear up some misunderstandings, and hope that our additional results can help to clarify some points about the scalability of CSM.
> > >
> > > With respect to the results from Appendix B, we agree that it remains a design choice as to how to construct the other kinds of graphs (e.g. determining the ordering of the dims for the cycle/chain). Some initial exploration of the data will be helpful for determining the best kind of neighborhood structure as in our short discussion in Appendix B.2.
> > >
> > > As for the exploding neighborhood sizes for high-dimensional cases, this is unfortunately an unavoidable phenomenon. However, such kinds of combinatorial intractability are quite common when dealing with discrete structures -- for example conducting inference over MRFs, structure learning from observational data, etc. Thus approximations (such as the one we used with the MNIST experiments) will be necessary for training more large-scale models. We believe that this research is important but will require a significant amount of additional theoretical and empirical work that we plan to conduct in the future. We hope that the reviewer will take into consideration that our current submission is a first step towards this goal by laying out the groundwork for this research.

---

### Official Review · Reviewer_oRLC · 2022-07-11

**Rating:** 8
**Confidence:** 4
**Soundness:** 3 good
**Presentation:** 3 good
**Contribution:** 3 good

**Summary:**

This work presents concrete score matching which generalizes score matching to discrete distributions. This is not the first work to attempt to generalize score matching to discrete domains. Lyu 2012, does this first, but defines a discrete ''score'' using the dimension-wise marginals of each dimension in the input. Conversely, here the ''score'' is created by defining a graph over states in the support of the distribution. The score is defined as the likelihood ratio between neighboring states in this graph. The authors provide sufficient conditions to ensure that learning the concrete score is equivalent to learning the desired distribution. This setup is more flexible than prior generalizations and can be applied to inputs which cannot be embedded into {1, ..., k}^D.

The authors present two learning methods taking advantage of the concrete score; concrete score matching and denoising concrete score matching -- generalizations of score matching and denoising score matching respectively. As well, they present an efficient training procedure for both and an MCMC sampling procedure which utilizes the concrete scores.

They present experimental results on some toy discrete distributions, non-toy discrete distributions and binary MNIST. The presented results in dictate that CSM performs favorably.

**Questions:**

While I enjoyed the work, I am left with quite a few questions.

Neighborhood structure: I know you mention that the neighborhood structure should be an important design choice. This makes sense. Did you perform any ablations on this? It would be really nice to see a likelihood based experiment like the section 5.4 using different neighborhoods. This would give the reader some intuition on how this should be chosen.

Section 5.4: I like that you used a normalized model to test these various approaches. This enables likelihood based evaluation which you used. I was somewhat curious as to why there was no maximum likelihood baseline presented in the table. I think this should be put in the table. No one would ever expect a method like this (which does not require a normalized model) to outperform ML training but it can provide a good ceiling on the performance of a model under ideal circumstances. This would greatly improve the impact of these results.

Efficient Training: In section 4.2 you present an efficient training scheme where you sample neighbors in the graph uniformly. Have you considered using a gradient-based importance sample scheme as in [1, 2]? I know that may be outside the scope of the work but I think it could be a nice extension that would be worth noting on here.

Denoising concrete score matching: I like that you propose two learning methods for the concrete score. Unfortunately it seems that you present no experiments using this method. It would be nice to present some result (even a toy result) indicating that this method, in practice, can be used to train a model.

Evaluation: Have you considered using something like AIS or RAISE to evaluate these models? I don't see why you couldn't do that to provide a bound on log-probability for your MNIST models, for example. Even if you don't present these results, it might be nice to add a note in section 3.3

[1] https://openreview.net/pdf?id=IEKL-OihqX0
[2] https://arxiv.org/abs/2102.04509

**Limitations:**

Yes, I believe the authors have adequately addressed the limitations of their work.


**Strengths And Weaknesses:**

Strengths:

I enjoyed reading this paper. I think the method is an elegant generalization of score matching and synthesizes prior attempts to generalize score matching to discrete domains. The method makes sense, is easy to understand, and seems to be well justified. The paper is not bogged down with unnecessary theory, but the theory presented is useful in helping the reader build understanding and intuition about the method.

The experimental results in the paper were nice. I liked the use of the MADE model for section 4.2 although I have some thoughts that I will elaborate on in the next section. The MNIST results are also quite nice. It is very difficult (even on MNIST) to obtain such nice results using (non-de noising) score matching, so I was quite impressed.


Weaknesses:

I think the main weaknesses with the paper are in the empirical study of the method. I think there are a number of simple things that could have been done which would have greatly improved this. For example, there could have been a study of the impact of the choice of neighborhood structure. As well, there could have been a maximum likelihood baseline presented in section 5.4. Further, while denoising concrete score matching was presented, there were no empirical results shown. All of these things could have pretty easily been added and would have greatly improved the quality of the results section.

---

> ### Author Response · Authors · 2022-08-01
> **Extra empirical analysis is added to the supplementary material.**
>
> **Q: MLE baseline in Table 1 (Section 5.4).**
>
> **A:** Thank you for this suggestion! We present additional results in Table 2 (Appendix B.1, Supplementary Material) with the appropriate discussion, where we trained a MADE model using standard maximum likelihood. As expected, this maximum likelihood baseline serves as an upper bound on performance across all score-based methods as it directly optimizes likelihood during training.
>
> We note that as in continuous (Stein) score matching, log-likelihoods and score matching losses are not always correlated even though they both theoretically converge to the optimal solution given infinite model capacity [20]. This is due to practical constraints such as model mis-specification or optimization challenges. We believe that this is also the case for discrete score matching, which explains why directly optimizing likelihood outperforms all other score-based approaches in Table 2.
>
> **Q: Practical selection of neighborhood structure.**
>
> **A:** We added Appendix B.2 in the revised PDF with additional experiments on synthetic datasets to elaborate upon this point. In particular, we experimented with 4 different neighborhood structures: a complete graph, as well as 3 different orderings of nodes in a cycle. In a setting where the data distribution is well-behaved (does not contain any low-density regions), we found that the neighborhood structure did not play a significant role in the model’s final performance as evaluated by log-likelihood—all structures performed similarly (Figure 5 in Appendix B.2). However, when we modified the data distribution to be more ill-conditioned (i.e., contain low-density regions between high density modes), a particular cycle configuration as well as the complete graph converged faster than all other approaches (Figure 6 in Appendix B.2). This observation shares similarity with Stein score optimization ([21] Song et al.), where low-density regions in the data distribution pose additional challenges to optimization and learning in practice (and in our case neighborhood selection).
>
> Such results emphasize that while the neighborhood structure can be any connected structure in theory (Theorem 1), the best-performing structure in practice is often data-dependent. That is, a neighborhood graph which respects/takes into account the particular dataset structure will perform the best (analogous to the way in particular types of inductive biases are useful for solving specific tasks). We believe that in practice—especially for high-dimensional datasets—selecting the right neighborhood structure is a crucial aspect of CSM, and additional heuristics may be needed to design the ideal structure. We have also added a short discussion about this in Appendix B.2.
>
>
> **Q: Denoising CSM.**
>
> **A:** We present additional results on the denoising variant of CSM (D-CSM) in Appendix B.3. We focus on 2-D synthetic datasets commonly used in the density estimation literature, and consider discrete categorical distributions with different variances for the noise distribution $\tilde q(\tilde x| x)$. As shown in Figure 7 (Appendix B.3), we observe that D-CSM is able to generate reasonable samples that match the ground truth distribution across various noise levels, demonstrating the promise of D-CSM. We leave a more in-depth exploration of this approach for future work.
>
>
> **Q: Additional references.**
>
> **A:** Thank you for the additional reference (Liu et al. 2022) – we were not aware of it as it seems to be a paper that is also under review at ICLR this year. Their approach is an interesting way to scale Ratio Matching in practice, and we have added this to our Related Works (Section 6). Leveraging importance sampling in the training objective as a variance reduction technique, or even as a way to potentially identify the best possible neighborhood structure of a given dataset, is very interesting future work, and we have added a discussion about this in Appendix B.2.
>
>
> **Q: AIS/RAISE evaluation.**
>
> **A:** This is a good point. We have added a discussion on evaluating the log-likelihood of our models with AIS/RAISE in Section 3.3.

---

> > ### Comment · Reviewer_oRLC · 2022-08-06
> > **Thanks!**
> >
> > I thank the authors for their responses to my comments and questions. I really appreciate the new experiments that they ran. I feel the new results really build out the picture they've painted and strengthen the work. I think the theory and methods presented here could be very useful in training large-scale models of discrete data if handled properly. For that reason I'm inclined to raise my score and advocate for this paper's acceptance.

---

> > > ### Author Response · Authors · 2022-08-06
> > > **Appreciate the response!**
> > >
> > > Thank you for your careful consideration of our response and additional results -- we appreciated the suggestions and agree that the updated paper is stronger.

---

### Author Response · Authors · 2022-08-01
**Summary**

We thank the reviewers for their insightful comments and suggestions! We appreciate the reviewers’ unanimous positive feedback in terms of our work’s **elegance, soundness of theoretical claims, and quality**. We summarize the answer to common concerns below, then address reviewer-specific comments separately. We have clarified the writing and included additional experiments (Appendix B) as requested in the revised PDF and appendix where our changes are marked in blue.

**Common questions:**

**Selection of neighborhood structure (R1, R2):** A common question posed by the reviewers was how the neighborhood structure should be selected in practice. We agree that although our theory suggests that the neighborhood structures can be any connected graph (Theorem 1), the best-performing structure empirically is often data-dependent. That is, a neighborhood graph which respects/takes into account the particular dataset structure will perform the best (analogous to the way in particular types of inductive biases are useful for solving specific tasks). To better understand this point, we add a new section (Appendix B.2) where we studied and experimented with different neighborhood structures on synthetic datasets.

**Scalability to higher dimensions (R2, R3):** We would like to first address R2’s point: Theorem 1 does not require a fully-connected graph (i.e., complete graph). As mentioned in Figure 1 (as well as in l. 98-106), we only require that the neighborhood structure is connected. This allows for the specification of a variety of graphs such as stars and chains. And although we use Monte Carlo approximations to approximate the sum over all neighbors in Theorem 3 for high-dimensional datasets, we provide efficient training objectives for the loss in Algorithms 1 and 2. This is exactly why it is feasible to use the neighborhood structure specified in l. 203 and 220 in the original text (l. 211 and 228 in the revised PDF) as pointed out by R3. In particular for our MNIST experiments, we found that such slicing/projection steps (sampling by neighbors) worked well.

---

### Public Comment · ~Yeongbin_Seo1 · 2025-01-16
**A few questions**

Thank you for great work! I have some questions here.

1. **Regarding Section 5.3 (Sampling with 2-D Multi-Class Dataset) and Figure 3**: How is this figure explained in terms of \(x\) and \(x_{nk}\)? For instance, does one pixel represent \(x\), with adjacent pixels being treated as \(x_{nk}\)?

2. **Regarding Sections 4.1 and 4.2 of the CSM Paper**: I’m curious about how \(c_\theta(x; N)\) is measured during training. Since \(c\) requires knowledge of both \(p(x)\) and \(p(x_{nk})\), how do you sample \(p(x_{nk})\) in practice?
    In Algorithm 1, it mentions uniformly sampling \(i\) within 1∼ K for a given \(x\). I guess this make all \(p(x_{nk})\) also have uniform probabilities.

---

### Meta-Review · Area_Chair_qz9n · 2022-08-26

**Recommendation:** Accept
**Confidence:** Certain

**Metareview:**

All reviewers are positive about the paper.

- The paper is well-written / sound and the experiments are convincing
- It's true that the neighborhood selection and lack of scalability w.r.t. dimension are limitations but they are acknowledged.
- Reviewers raised valid concerns but the rebuttal addressed many of them.

Overall, the formulation presented here is quite general and could have a great deal of application and impact especially given the explosion of work on score-based models in continuous domains.

We recommend acceptance.

**Award:**

No

---

### Decision · Program_Chairs · 2022-09-14

Accept